# Learning visual-linguistic adequacy, fidelity, and fluency for novel object captioning

## Abstract

Novel object captioning (NOC) learns image captioning models for describing objects or visual concepts which are unseen (i.e., *novel*) in the training captions. Such captioning models need to sufficiently describe such visual data with fluent and natural language expression. In other words, we expect the produced captions being linguistically fluent, containing novel objects of interest, and fitting the visual concept of the image. The above three aspects thus correspond to *fluency*, *fidelity*, and *adequacy*, respectively. However, most novel object captioning models are not explicitly designed to address the aforementioned properties due to the absence of caption annotations. In this paper, we start by providing an insight into the relationship between the above properties and existing visual/language models. Then, we present *VLAF2*, for learning Visual-Linguistic Adequacy, Fidelity, and Fluency, which utilizes linguistics observed from captions for describing visual information of images with novel objects. More specifically, we revisit BERT and CLIP, and explain how we leverage the intrinsic language knowledge from such popular models to reward captions with precise and rich visual content associated with novel images. To validate the effectiveness of our framework, we conduct extensive experiments on the nocaps dataset. Our method not only performs favorably against state-of-the-art novel captioning models in all caption evaluation metrics, but also surpasses the SPICE scores of human baseline. We perform quantitative and qualitative analysis to demonstrate how our model generates novel object captions with improved fluency, fidelity, and adequacy. Implementation details and code are available in the supplementary materials.

## 1 Introduction

Novel Object Captioning (NOC) (Agrawal et al., 2019) requires captioning models to accurately capture images containing novel objects unseen during training captions, and to describe such data with fluent and grammatically correct sentences. Despite impressive benchmark performance on COCO Captions (Chen et al., 2015) and Flickr (Young et al., 2014), existing image captioning models (Gao et al., 2019; Huang et al., 2019; Wang et al., 2019; Guo et al., 2020; Pan et al., 2020; Cornia et al., 2020; Zhou et al., 2020) or unpaired image captioning (Gu et al., 2019; Feng et al., 2019) generalize poorly to this task since the NOC task covers a larger variety of visual concepts, where nearly 400 object classes barely have any associated training captions.

Existing work of novel object captioning typically rely on the object detection results to either fill in the generated slotted sentences (Lu et al., 2018; Wu et al., 2018) or learn the visual vocabulary for novel objects directly (Hu et al., 2020). However, these methods do not consider the semantics and linguistics of the entire sentences comprehensively. Specifically, most existing works do not exhibit the abilities in assuring the produced captions with correct and rich visual content, or with sufficient natural and fluent expression. In other words, the three aspects or challenges of NOC still need to be addressed. First of all, *fluency* is expected in the output caption which is linguistically natural and grammatically correct. *Fidelity* reflects the novel objects to be described, while *adequacy* encourages output captions to properly capture the visual concept of the input image.

Unfortunately, the above properties cannot be easily achieved by standard image captioning models due to the absence of ground truth caption annotations. In this paper, we revisit two popular pre-trained visual/language models of BERT (Devlin et al., 2018) and CLIP (Radford et al., 2021),

Figure 1: Overview of VLAF2 captioning model. (a) Leveraging linguistic ability from BERT for describing novel objects. (b) Preserving adequacy and fidelity of novel object captions via CLIP.

and we provide the associated connection to fluency, fidelity, and adequacy for NOC. As shown in Fig. 1, we present *VLAF2* for learning Visual-Linguistic Adequacy, Fidelity, and Adequacy, which leverages the intrinsic knowledge of BERT and CLIP for utilizing linguistics observed from captions for describing visual information of images with novel objects. With BERT pre-trained to excel at various linguistic tasks and CLIP to associate visual-language data at instance levels, we provide insights to these models and present technical details on how such models can be utilized for performing NOC, while the goals of adequacy, fidelity and fluency can be jointly achieved.

For the evaluation part of our work, we conduct extensive experiments on benchmark nocaps datasets, confirming that our model is able to produce state-of-the-art results in terms of metrics linking to adequacy, fidelity, and fluency. In addition, by a variety of ablation studies, we further provide analysis on how BERT and CLIP are utilized and thus be preferable in solving NOC tasks.

## 2 CONNECTING FLUENCY, FIDELITY AND ADEQUACY WITH BERT AND CLIP

We first discuss how fluency, fidelity, and adequacy can be fundamentally and technically related to the visual/language models of BERT (Devlin et al., 2018) and CLIP (Radford et al., 2021), which will be utilized in our proposed learning framework. For caption fluency, one expects that the image caption output to be linguistically natural and grammatically correct. It not only requires to capture the co-occurrence of novel-object vocabularies, the associated collocations such as verbs or modifiers are expected to be properly utilized. We define the context containing novel-object vocabularies as $\tilde{y}$ and the associated collocations as $\hat{y}$. Fluency can be defined as how many collocations are observed by captioning models given a novel image context $p(\hat{y}|\tilde{y})$. This is exactly the masked language modeling (MLM) objective adopted in BERT, without the log function explicitly calculated. This is the reason why we adopt BERT to learn the co-occurrence of novel-object vocabularies and their associated collocations to improve the linguistic fluency.

We now relate fidelity and adequacy in image caption outputs to the model of CLIP. We start by defining objects appearing in the caption as $\mathcal{X}$, and objects mentioned by the captions as $\mathcal{Y}$. Relevant objects $(\mathcal{X}, \mathcal{Y})$ are defined as objects that are both included in the images and described by the captions. Fidelity assesses whether the visual content presented in the produced caption is correct, and can be defined as the fraction of relevant objects among objects in captions $p(\mathcal{X}|\mathcal{Y}) = \frac{p(\mathcal{X}, \mathcal{Y})}{p(\mathcal{Y})}$. Similarly, adequacy evaluates whether sufficient visual details have been expressed by captions, and we can define it as the fraction of relevant objects among objects in images $p(\mathcal{Y}|\mathcal{X}) = \frac{p(\mathcal{X}, \mathcal{Y})}{p(\mathcal{X})}$. Based on the observation of Friedman (2017), these two quantities $p(\mathcal{X}|\mathcal{Y})$ and $p(\mathcal{Y}|\mathcal{X})$ represent the association between $\mathcal{X}$ and $\mathcal{Y}$. While CLIP is trained to associate image and text data at the instance level, it can be further applied for guiding image captioning models for NOC, boosting the desirable fidelity and adequacy.

## 3 METHOD

Before presenting the learning framework of VLAF2 for novel object captioning, we determine the notations and settings for the sake of completeness. Given a small set of caption-labeled images $X_l$, the corresponding captions $Y_l$, as well as a large set of uncaptioned images $X_u$, our goal is to generate the associated captions $\hat{Y}_u$ for $X_u$ using the captioning model $C_\theta$, where $\theta$ is the parameters of captioning model $C$. To achieve this, we propose a two-stage learning framework of VLAF2, guided by two pre-trained visual-language models of BERT (Devlin et al., 2018) and CLIP (Radford

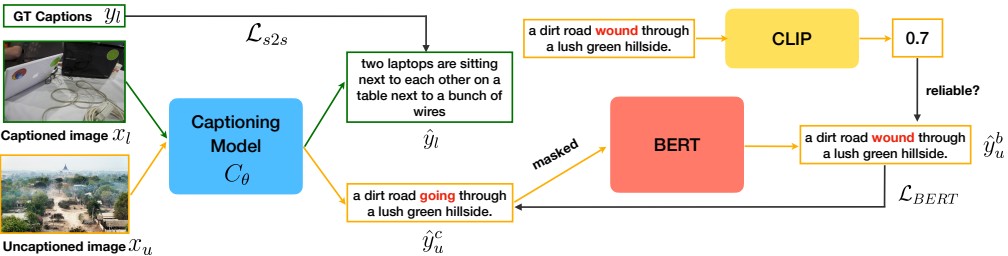

Figure 2: Learning to caption novel objects with linguistic fluency. For caption-labeled image $x_l$, we impose the sequence-to-sequence objective $\mathcal{L}_{s2s}$ for training. For uncaptioned image $x_u$, we exploit BERT to improve the wordings of the generated caption $\hat{y}_u^c$, and the refined caption is denoted as $\hat{y}_u^b$.

et al., 2021). Note that neither BERT nor CLIP have observed $X_u$ during training. An overview of our method is shown in Fig. 1. While our captioning model determines the labels of novel object images using pretrained VIVO (Hu et al., 2020), our major technical contribution lies in how we leverage visual-linguistic information from BERT to CLIP, realizing the goal of producing novel object captions with sufficient fluency, fidelity and adequacy.

Take Fig. 1(a) for example, when the captioning model generates a caption for a given image, VLAF2 would regularize and substitute the verb *wound* for *going*, ensuring the corresponding linguistic fluency. Followed by next training stage in which we randomly samples multiple captions from the same image with reward captions properly designed, VLAF2 would further produce captions with improved fidelity and adequacy. For example, in Fig. 1(b), the caption in the lower box accurately describes the object *red scarf* and thus receives a higher reward from CLIP. This encourages the model to correctly describe the visual content and capture the associated visual concept of the input image. How the pre-trained models of BERT and CLIP would guide the learning of our VLAF2 will be detailed in the following subsections.

## 3.1 DESCRIBING NOVEL OBJECTS WITH LINGUISTIC FLUENCY

By observing image-caption pairs $(x_l, y_l)$, the image captioning model in Fig. 1(a) would learn the visual grounding (i.e., localization of known objects and referring their expressions) as well the linguistics of captions. With uncaptioned images $X_u$ containing novel objects, we adopt BERT to refine the wordings for captions containing novel objects, followed by CLIP to assess the quality of the resulting caption outputs.

While initialized by pre-trained models of VIVO (Hu et al., 2020), our captioning model in Fig. 1(a) would recognize images with novel objects but lack sufficient ability in "describing" them in terms of captions. To solve this problem, we impose a conventional sequence-to-sequence objective $\mathcal{L}_{s2s}$, which requires supervision of image-caption pairs $(x_l, y_l)$. That is, we have

$$\mathcal{L}_{s2s} = \text{CrossEntropy}(\hat{y}_l, y_l), \tag{1}$$

where $\hat{y}_l = C_\theta(x_l)$ denotes the predicted caption. As for uncaptioned images $X_u$, while we do not observe ground truth caption for images containing novel objects, the collocations of the associated novel objects would be discovered via exploiting the intrinsic knowledge of BERT. Thus, the linguistic fluency of resulting captions can be further improved.

We now detail the learning process for the aforementioned uncaptioned image data. As illustrated in Fig. 1, given an uncaptioned image $x_u$, the captioning model $C_\theta$ generates a caption $\hat{y}_u^c = C_\theta(x_u)$, $\hat{y}_u^c = \{w_1^c, w_2^c, ..., w_T^c\}$, where $w_i^c$ denotes the $i$th word and $T$ is the caption length. The superscript $c$ represents it is generated by our captioning model. We then obtain a masked caption $\hat{y}_u^m = \{w_1^c, w_2^c, ..., w_M^m, ..., w_T^c\}$ with $m$ indicating the mask index, by randomly masking out the words in the caption. We note that, we do not mask nouns in the above process, since they are viewed as relating to objects grounded in the visual content, and language models like BERT are not designed to handle such information. Finally, BERT takes the masked sequence $\hat{y}_u^m$ as input and recovers the masked word conditioned on the semantics of the entire sentence, producing the refined caption $\hat{y}_u^b = \text{BERT}(\hat{y}_u^m)$, $\hat{y}_u^b = \{w_1^c, w_2^c, ..., w_M^b, ..., w_T^c\}$.

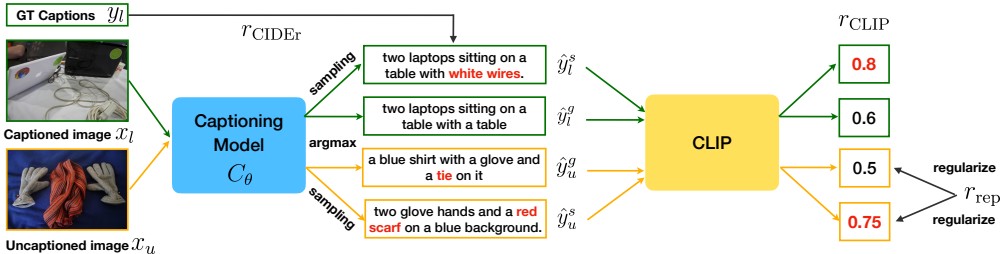

Figure 3: Learning to caption novel objects with improved visual-linguistic adequacy and fidelity. For caption-labeled image $x_l$, we perform SCST training using CIDEr as reward. The sampled caption $\hat{y}_d^s$ will be rewarded by CLIP if it has higher cross-modal association than the greedy-decoded baseline $\hat{y}_d^g$. The superscript $d$ indicates the source of the image. Additionally, we regularize our model with $r_{\text{rep}}$ to avoid redundant and repetitive captions.

It is worth pointing out that, however, not every word substitution from BERT is semantically correct. This is the reason why we require CLIP to validate each replacement output. Specifically, if the refined caption comprises a more accurate and associated word that human generally uses to describe the scene, then a higher CLIP score would be obtained (than that of the original caption). Thus, we propose the objective function $\mathcal{L}_{BERT}$ for learning wordings, which calculates the cross-entropy loss for the replaced words, gated by the comparison of the CLIP scores of the two captions. More precisely, $\mathcal{L}_{BERT}$ is derived as:

$$\mathcal{L}_{BERT} = g \cdot \text{CrossEntropy}(w_M^c, w_M^b), \quad g = \begin{cases} 0 & \text{if } \text{CLIP}(x_u, y_u^b) \leq \text{CLIP}(x_u, y_u^c) \\ 1 & \text{if } \text{CLIP}(x_u, y_u^b) > \text{CLIP}(x_u, y_u^c) \end{cases}, \quad (2)$$

where $\text{CLIP}(x, y)$ calculates the association between an image $x$ and its caption $y$, indicating how well the captions match the images. More details can be referred to Radford et al. (2021).

## 3.2 LEARNING NOVEL OBJECT CAPTIONS WITH FIDELITY AND ADEQUACY

Preserving the adequacy and fidelity of captions for images with novel objects is another challenging task. Recall that, as discussed in Sec. 2, fidelity verifies the correctness of visual content presented in the generated caption, while adequacy assesses whether sufficient visual details have been expressed in it. Conventional sequence-to-sequence model training with cross-entropy loss might not reflect the above desirable properties. This is because that, models trained with word-by-word supervision tend to imitate the sentence patterns of the training images instead of relating caption data to the visual content, leading to less optimal caption generation.

To tackle the above issues, we utilize the captioning evaluation metric of CIDEr as the reward in our learning framework, since it encourages the generated caption to be consistent with that of the human annotated ones in the word level. However, while CIDEr can be easily computed for captioning labeled images $X_l$, it cannot be explicitly calculated for captioning uncaptioned images $X_u$ due to the absence of ground-truth captions. Instead, we propose to optimize for fidelity and adequacy of the generated captions via the association between images and captions calculated by CLIP, encouraging captions which precisely describe the objects with plentiful visual details, as discussed in Sec. 2. However, we observe that captioning models would achieve improved association by simply repeating the same object that occurs in the image, which undermines the linguistic fluency of the captions. For example, the image caption *"a group of cans of soda and other items on a table"* can be replaced by *"a pile of **cans** and bottles of soda on a counter with **cans** of **cans**"* with a higher CLIP score. Therefore, we additionally impose a repetition penalty to avoid such trivial solutions. We now describe how the rewards are calculated and how do we update the captioning models.

**Rewards for the generated captions.** In order to realize the above properties, we design the following rewards to reflect the quality of the generated captions. For image-caption pairs $(x_l, y_l)$, we directly calculate the CIDEr reward for the predicted caption $\hat{y}_l$ (i.e., $r_{\text{CIDEr}} = \text{CIDEr}(\hat{y}_l, y_l)$). To encourage the generated captions for $X = X_l \cup X_u$ with fidelity and adequacy, we exploit CLIP to compute the association between $X = X_l \cup X_u$ and $\hat{Y} = \hat{Y}_l \cup \hat{Y}_u$ (i.e., $r_{\text{CLIP}} = \text{CLIP}(x, \hat{y})$).

As for repetition penalty to preserve linguistic fluency of the generated captions $\hat{y}_u = \{w1, w2, ..., w_T\}$ for $x_u$, we formulate it as a linear assignment problem, where every word is assigned to the most similar one in the sentence except for itself. Then, we calculate the similarity between such pairs for each sentence. Intuitively, repetitive captions would have high similarity scores, since the repeated words will be assigned to the exact same words. We define the assignment $\hat{\alpha}$ as the one that maximizes the average pairwise similarity of a sentence. Thus, we have:

$$\hat{\alpha} = \arg\max_{\alpha} \frac{1}{T} \sum_{i=1}^{T} C(w_i, w_{\alpha(i)}), \tag{3}$$

where $\alpha(i)$ is the index of the word assigned to the $i$-th word in the caption, and $C(w_i, w_j)$ is the cosine similarity between the GloVe (Pennington et al., 2014) word representation of two words. Since a desirable captioning model would encourage captions with low repetition (i.e., low average pairwise similarity), the reward for repetition penalty is defined as follows:

$$r_{\text{rep}} = 1 - \frac{1}{T} \sum_{i=1}^{T} C(w_i, w_{\hat{\alpha}(i)}), \tag{4}$$

Note that we do not calculate repetition penalty for $\hat{Y}_l$ since they are regularized by the aforementioned CIDEr rewards. With the above discussions, the total reward for caption-labeled data would be $r(\hat{y}_l) = r_{\text{CIDEr}}(\hat{y}_l, y_l) + r_{\text{CLIP}}(x_l, \hat{y}_l)$, and the total reward for uncaptioned data would be $r(\hat{y}_u) = r_{\text{CLIP}}(x_u, \hat{y}_u) + r_{\text{rep}}(\hat{y}_u)$.

**Back-propagation via reinforce algorithm.** Unfortunately, computation of the aforementioned rewards is non-differentiable. Thus, we adopt reinforce algorithm (Williams, 1992) to optimize the learning of our model. As shown in Fig. 1, for an image $x$ we randomly sample the caption $\hat{y}^s$ from the word distribution and use greedy decoding to obtain the baseline result $\hat{y}^g$. If the sampled captions possess higher linguistic fluency or cross-modal association than the baseline caption, they will be encouraged by positive rewards and vice versa. We follow Rennie et al. (2017); Liu et al. (2017; 2018) and define the objective function as follows:

$$\nabla_{\theta} \mathcal{L}_{RL}(\theta) \approx -(r(\hat{y}_d^s) - r(\hat{y}_d^g)) \nabla_{\theta} \log p_{\theta}(\hat{y}_d^s),$$

$$r(\hat{y}_d) = \begin{cases} (r_{\text{CIDEr}}(\hat{y}_d, y_d) + r_{\text{CLIP}}(x_d, \hat{y}_d) & \text{if } x_d \in X_l \\ r_{\text{CLIP}}(x_d, \hat{y}_d) + r_{\text{rep}}(\hat{y}_d) & \text{if } x_d \in X_u \end{cases}, \tag{5}$$

where $d$ indicates the source of the image, $\theta$ being the parameters of captioning model, and $p_{\theta}(\hat{y}^s)$ represents the predicted word logits for the generated captions. With the objective functions defined in equations (1), (2), and (5), our captioning model $C_{\theta}$ can be trained accordingly.

## 4 EXPERIMENT

**Datasets & Implementation Details.** The training data for the **nocaps** benchmark comprises the Open Images V4 (Kuznetsova et al., 2020) object detection training set (1.7M images annotated with bounding boxes for 600 object classes), plus the image-caption pairs from the COCO Captions 2017 (Chen et al., 2015) training set (118K images containing 80 object classes). No additional image-caption pairs are provided for training. We refer the images from the Open Images dataset to the uncaptioned images $X_u$ we define in Sec. 3, and the image-caption pairs from COCO Captions are defined as $(X_l, Y_l)$. We only use the Open Images datasets during VIVO pre-training but leverage both datasets for training as described in Sections 3.1 and 3.2. We evaluate our model on the validation and test set of nocaps, which comprises 4500 and 10600 images from the Open Images validation and test sets, respectively. For the architecture of our captioning model, we follow (Hu et al., 2020; Li et al., 2020; Zhang et al., 2021) to use a BERT-base (Devlin et al., 2018) model. As for CLIP and BERT, we directly exploit the pre-trained models released by their authors. The architecture for BERT is BERT-Large, and the version for CLIP is ViT/B-32. Due to page limits, hyperparameters and other training details can be found in the Appendix A.

### 4.1 EVALUATION METRICS

**CIDEr.** Similar to evaluation metrics (Papineni et al., 2002; Lin, 2004; Banerjee & Lavie, 2005) for NLP tasks, Consensus-based Image Description Evaluation (CIDEr) calculates the similarity

Table 1: Quantitative results on nocaps. The numbers before/after slashes denote scores derived without/with Constrained Beam Search (CBS).

| Method | in-domain | | near-domain | | out-of-domain | | overall | |
|---|---|---|---|---|---|---|---|---|
| | CIDEr | SPICE | CIDEr | SPICE | CIDEr | SPICE | CIDEr | SPICE |
| Validation Set | | | | | | | | |
| UpDown | - / 79.3 | - / 12.4 | - / 73.8 | - / 11.4 | - / 71.7 | - / 9.9 | - / 74.3 | - / 11.2 |
| Oscar$_B$ | - / 83.4 | - / 12.0 | - / 81.6 | - / 12.0 | - / 77.6 | - / 10.6 | - / 81.1 | - / 11.7 |
| Oscar$_L$ | - / 85.4 | - / 11.9 | - / 84.0 | - / 11.7 | - / 80.3 | - / 10.0 | - / 83.4 | - / 11.4 |
| Oscar$_B$ + VIVO | - / 92.2 | - / 12.9 | - / 87.8 | - / 12.6 | - / 87.5 | - / 11.5 | - / 88.3 | - / 12.4 |
| VinVL | 97.9 / 96.8 | 13.2 / 13.5 | 89.2 / 90.7 | 12.9 / 13.1 | 68.4 / 87.4 | 10.8 / 11.6 | 86.1 / 90.9 | 12.5 / 12.8 |
| VinVL + VIVO | 95.8 / 94.8 | 13.3 / 13.3 | 90.5 / 91.4 | 12.8 / 13.0 | 77.1 / 88.7 | 11.1 / 11.6 | 88.6 / 91.4 | 12.5 / 12.7 |
| Human | 84.4 | 14.3 | 85.0 | 14.3 | **95.7** | **14.0** | 87.1 | **14.2** |
| **Ours** | **102.8** / **101.4** | **14.8** / **15.1** | **97.9** / 96.8 | **14.4** / **14.5** | 86.3 / **95.4** | 12.5 / 12.9 | 96.3 / **97.2** | 14.1 / **14.2** |
| Test Set | | | | | | | | |
| UpDown | - / 76.0 | - / 11.8 | - / 74.2 | - / 11.5 | - / 66.7 | - / 9.7 | - / 73.1 | - / 11.2 |
| Oscar$_B$ | - / 81.3 | - / 11.9 | - / 79.6 | - / 11.9 | - / 73.6 | - / 10.6 | - / 78.8 | - / 11.7 |
| Oscar$_L$ | - / 84.8 | - / 12.1 | - / 82.1 | - / 11.5 | - / 73.8 | - / 9.7 | - / 80.9 | - / 11.3 |
| Oscar$_B$ + VIVO | - / 89.0 | - / 12.9 | - / 87.8 | - / 12.6 | - / 80.1 | - / 11.1 | - / 86.6 | - / 12.4 |
| VinVL | 93.0 / 93.8 | 13.3 / 13.3 | 84.7 / 89.0 | 12.7 / 12.7 | 64.0 / 66.1 | 10.9 / 10.9 | 82.0 / 85.5 | 12.4 / 12.5 |
| VinVL + VIVO | **94.5** / 84.4 | 13.1 / 12.8 | 90.9 / 86.0 | 12.9 / 12.6 | 73.9 / 77.9 | 11.2 / 11.3 | 88.3 / 84.4 | 12.6 / 12.4 |
| Human | 80.6 | **15.0** | 84.6 | **14.7** | **91.6** | **14.2** | 85.3 | **14.6** |
| **Ours** | **101.7** / 90.4 | **15.0** / 14.3 | **95.7** / 91.0 | **14.4** / 14.0 | 78.9 / **82.5** | 12.1 / **12.2** | **93.5** / 89.4 | **14.1** / 13.7 |

Table 2: Quantitative comparisons on caption fluency, fidelity and adequacy. Note that BLEU@4 (B@4) and CIDEr (C) are utilize for describing fluency, object precision (P) for fidelity, object recall (R) for adequacy and object F1 scores (F1) for overall cross-modal association.

| Method | in-domain | | | | | near-domain | | | | | out-of-domain | | | | |
|---|---|---|---|---|---|---|---|---|---|---|---|---|---|---|---|
| | B@4 | C | P | R | F1 | B@4 | C | P | R | F1 | B@4 | C | P | R | F1 |
| VinVL | 32.6 | 63.2 | **59.2** | 40.8 | 48.3 | 30.5 | 58.3 | 22.8 | 32.6 | 26.8 | 29.6 | 48.8 | 48.4 | 25.6 | 33.5 |
| VinVL+VIVO | 31.2 | 59.8 | 56.0 | 42.2 | 48.1 | 30.1 | 57.3 | 28.5 | 36.3 | 32 .0 | 27.3 | 45.6 | 49.0 | 27.3 | 35.1 |
| **Ours** | **35.9** | **68.0** | 58.2 | **45.6** | **51.3** | **32.2** | **60.8** | **39.9** | **41.0** | **40.4** | **30.1** | **50.2** | **51.3** | **30.5** | **38.3** |

between the reference and generated caption by word n-gram overlap in a rule-based manner. To capture human consensus for image captioning evaluation, it introduces the tf-idf weight to reduce the matching weight of the n-grams that are common in all image captions.

**SPICE.** Semantic Propositional Image Caption Evaluation (SPICE) (Anderson et al., 2016b) matches the semantics between sentences, such as objects, relations, and attributes of objects. Specifically, it converts sentences into semantic scene graphs, which allows evaluation to break grammatical constraints and focuses on propositional semantic content. It reflects the accuracy of the visual content and considers less about linguistic properties.

**Fluency.** To quantitatively evaluate fluency, we remove the effect of the visual information and focus on the quality of linguistic properties in the conventional caption evaluation metrics. Specifically, we remove all the objects and nouns from the captions and report BLEU@4 (Papineni et al., 2002) and CIDEr scores calculated by the removed version of reference and candidate captions. Note that the fluency experiment is conducted on a subset of the nocaps validation set, which contains 1000 images whose caption annotations are available on the official website of the nocaps dataset.

**Fidelity & Adequacy.** Fidelity and adequacy evaluate how well the captions are associated with images. As defined in Sec. 2, fidelity stands for the fraction of relevant objects described in captions among all the objects in captions, and adequacy is the fraction of relevant instances that were retrieved. These two properties are analogous to the definition of precision and recall, respectively. Therefore, we extract the objects mentioned in the captions and the ground-truth objects in the images and calculate the precision (for fidelity), recall (for adequacy), and F1 (for overall association) scores. The experiment is performed on the validation set of nocaps.

Following Agrawal et al. (2019), we further split the dataset into three subsets for evaluation: *in-domain* images only contain the seen objects that have been described in the training captions, *out-of-domain* images only with unseen (i.e., novel) objects presented, and *near-domain* ones containing both seen and unseen objects.

Table 3: Analyses on BERT, CLIP, and repetition penalty for NOC using nocaps validation set. Note that BERT mainly benefits the linguistic fluency with improved CIDEr, and CLIP is desirable for preserving visual semantics with increased SPICE.

| Method | in-domain | | near-domain | | out-of-domain | | overall | |
|---|---|---|---|---|---|---|---|---|
| | CIDEr | SPICE | CIDEr | SPICE | CIDEr | SPICE | CIDEr | SPICE |
| **Ours** | **102.77** | **14.83** | **97.90** | **14.40** | **86.33** | **12.54** | **96.25** | **14.10** |
| Ours w/o $\mathcal{L}_{BERT}$ | 99.13 | 14.41 | 94.72 | 14.11 | 84.47 | 12.38 | 93.27 | 13.81 |
| Ours w/o $r_{\text{CLIP}}$ | 101.12 | 13.80 | 94.07 | 13.35 | 80.54 | 11.94 | 92.33 | 13.14 |
| Ours w/o $r_{\text{rep}}$ | 96.73 | **14.83** | 89.64 | 14.12 | 81.87 | 12.38 | 89.08 | 13.88 |

Table 4: Analyses on BERT and CLIP for improving caption fluency, fidelity and adequacy. Note that BERT benefits fluency metrics of BLEU@4 (B@4) and CIDEr (C), while CLIP focusing on cross-modal association boosts metrics of object precision (P), recall (R), and F1 scores (F1).

| Method | in-domain | | | | | near-domain | | | | | out-of-domain | | | | |
|---|---|---|---|---|---|---|---|---|---|---|---|---|---|---|---|
| | B@4 | C | P | R | F1 | B@4 | C | P | R | F1 | B@4 | C | P | R | F1 |
| **Ours** | **35.9** | **68.0** | 58.2 | **45.6** | **51.3** | **32.2** | **60.8** | 39.9 | **41.0** | **40.4** | **30.1** | **50.2** | 51.3 | **30.5** | **38.3** |
| Ours w/o $\mathcal{L}_{BERT}$ | 32.8 | 64.8 | 58.1 | 40.9 | 48 | 30.4 | 60.6 | **41** | 39.0 | 40.0 | 27.9 | 49.1 | **51.6** | 27.9 | 36.2 |
| Ours w/o $r_{\text{CLIP}}$ | 33.2 | 65.9 | **58.8** | 42.1 | 49.1 | 31.9 | 60.7 | 35.8 | 37.7 | 36.8 | 28.7 | 49.8 | **51.6** | 27.4 | 35.8 |

## 4.2 QUANTITATIVE ANALYSIS

For performance comparisons, we choose UpDown (Agrawal et al., 2019) without SCST optimization (Rennie et al., 2017) and Oscar (Li et al., 2020) as baselines, as well as VinVL (Zhang et al., 2021) that achieves SOTA results on the benchmark of nocaps. In addition, VIVO (Hu et al., 2020) is a pre-training technique for captioning models, allowing them to recognize the novel objects. Since VinVL did not report the numbers with Constrained Beam Search (CBS) exploited during inference (CBS is known to improve model performance on out-of-domain data), we reproduce VinVL following details stated in the original paper. For more details please refer to Appendix A. For a comprehensive comparison, we conduct experiments on the validation and test set of nocaps. In addition, we compare our method with VinVL in fluency, fidelity, and adequacy to demonstrate the improvement in terms of these properties. We also evaluate on the COCO Caption dataset and report in Appendix B.

**The nocaps datasets.** The results on nocaps are shown in Table 1. From this table, we see that our model performed favorably against baselines and SOTAs across different metrics. We observe that CBS slightly decreased model performance on *seen* object captioning, since it forces captioning models to describe the *detected* objects without considering detection correctness. Nevertheless, for near or out-of-domain images, CBS still benefits the captioning performances. It is worth noting that our model largely increased the performance in SPICE score for every data domain, which verifies that our method is able to generate captions with the improved image-language association.

**Fluency, fidelity, and adequacy.** As described in Sec. 4.1, we design additional experiments for evaluating fluency, fidelity, and adequacy and report the numbers in Table 2. For fluency, we remove all the objects and nouns in the captions since they relate less to the linguistics of the captions. We then calculate the BLEU@4 (B@4) and CIDEr (C) scores for the captions after removal. For fidelity and adequacy, they indicate that captions should accurately (high precision) describe sufficient (high recall) visual details. Therefore, we report the object precision and recall in this table, and object F1 scores represent the overall association between captions and images. One can see that our method surpass previous methods by a visible margin on all tasks except for in-domain object precision., which further verifies our model improves novel object captioning on fluency, fidelity, and adequacy.

## 4.3 REMARKS ON BERT AND CLIP FOR CAPTION FIDELITY, ADEQUACY AND FLUENCY

We conduct ablation studies on the nocaps validation set, with the aim to verify the necessity of integrating BERT and CLIP for learning NOC models. Following the same evaluation procedures described in Sec. 4.1, we discuss the contributions of these models in terms linguistic and semantic level metrics in Tables 3 and 4. Detailed ablation analysis of every objective can be further found in Appendix B.2.

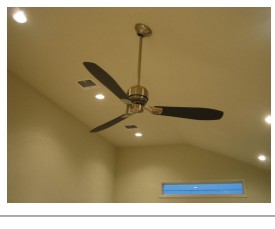 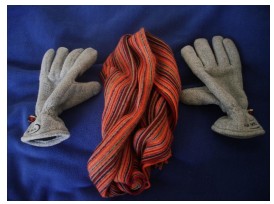 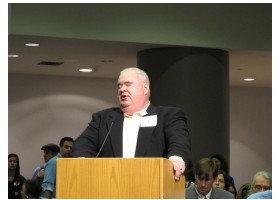

| | Fluency | Fidelity | Adequacy |
|---|---|---|---|
| VinVL | a ceiling fan **in** a room with a window | a **blue blanket** and **a glove** on a blue shirt. | a man in a suit and tie standing in a room. |
| Ours | a ceiling fan **hanging from** the ceiling in a room. | **two glove hands** and a **red scarf** on a blue background. | a man in a suit and tie standing at a **podium** in a room. |

| | Fluency | Fidelity | Adequacy |
|---|---|---|---|
| VinVL | a group of women **standing** on a field with a football. | a woman **sitting at a tennis table** with a racket. | a man and a woman playing an accordion. |
| Ours | a group of girls in football uniforms **posing** for a picture. | a woman in a **wheelchair** playing tennis at a table. | a couple of people **sitting in a church** playing an accordion. |

Figure 4: Example results and comparisons for image captions produced by VinVL and ours in terms of fluency, fidelity and adequacy. Note that both utilize VIVO for novel object detection.

**BERT.** As shown in Table 3, the captioning model without BERT would observe a performance drop in CIDEr for linguistic fluency, but such drops for the metric of SPICE (related to visual content) would be less significant. Similarly, as observed in Table 4, removing BERT would result in the lowest BLEU and CIDEr scores. These results confirm our motivation and model design discussed in Sec. 2, since BERT is utilized to improve caption quality at the linguistics level.

**CLIP.** As shown in Table 3, the model without CLIP showed significant drops in captioning metrics of CIDEr and SPICE. However, the performance decrease in SPICE is expected, since CLIP is particularly deployed in our framework for preserving visual content in captions. As for CIDEr, its decrease is mainly due to the deterioration of missing visual content in captions. This is also confirmed by Table 4, in which mainly the metrics reflecting fidelity and adequacy (i.e., at the visual semantics level) would observe significant drops for model trained without CLIP rewards.

**Repetition penalty.** Recall that, in Sec. 3.2, this penalty is to alleviate association between images and captions with redundant visual information. As seen in Table 3, the model without this penalty observed a significant performance drop in CIDEr scores. We did not see such trends for SPICE. This is because that, repetitive words in captions mainly violate linguistic structures rather then semantic accuracy, and thus the performance related to linguistic fluency would be more sensitive to the deployment of this penalty.

## 4.4 QUALITATIVE ANALYSIS

We now empirically show captions in Fig. 4, which are generated by our model and VinVL, with both pretrained from VIVO for novel object detection. In this figure, wordings that are less accurate or incorrectly describe the associated visual content are marked in bold. And, our wording improvements are highlighted in red. From this figure, one can see that for fluency, our model generated

vivid captions with more proper wordings. Take the upper-left image for example, our model particularly described *"fan hanging from the ceiling in a room"* instead of *"fan in a room"*. As for fidelity, our model is designed to capture the visual content in an image. Specifically, take the second image in the first row for example, we correctly described the number of gloves and the novel object *red scarf*, while VinVL failed to do so. As for adequacy, take the bottom-right image for example, our model was able to recover visual details in the image (i.e., *"people playing the accordion"* and *"sitting in a church"*). For more qualitative examples, please refer to Appendix B.5.

## 5 RELATED WORK

**Image captioning.** Recent progress of image captioning focuses on different model architectures and learning methods. Gao et al. (2019); Huang et al. (2019); Wang et al. (2019); Guo et al. (2020); Pan et al. (2020); Cornia et al. (2020) design different attention mechanisms for image captioning. Rennie et al. (2017); Li et al. (2019); Yang et al. (2020) adopt reinforcement learning to improve the performance. On the other hand, some researchers consider more challenging settings, such as partially supervised (Liu et al., 2018; Kim et al., 2019) or unpaired image captioning (Gu et al., 2019; Feng et al., 2019). However, these methods are restricted to the assumption that the unpaired images and captions share the same set of object class, and the number of object class is limited as well, which make them inapplicable to our task.

**Novel object captioning.** Previously, novel object captioning approaches (Anderson et al., 2016a; 2018; Hendricks et al., 2016) were only tested on a restrictive dataset with only eight novel object classes held out from the COCO dataset. Their extensions to large-scale image data with various novel objects are not sufficiently studied. Recent studies mainly rely on object detection results to improve the performance on novel object captioning. Lu et al. (2018); Wu et al. (2018) generate slotted caption templates, which are later filled in with visual concepts identified by object detectors. Yao et al. (2017) exploits a copying mechanism to assemble words corresponding to object detector predictions to generate captions. Similarly, Constrained Beam Search (CBS) (Anderson et al., 2016a) is an architecture-agnostic decoding algorithm that can be exploited during inference to enforce the inclusion of novel object classes in the captions. Instead of explicitly using detection results, Hu et al. (2020) learns the relationship between image and text by aligning object detection tags with their corresponding image region features. Recently, Wang et al. (2021a) indicate that a desirable caption should comprise properties of fluency, fidelity, and adequacy. Nevertheless, most existing NOC approaches are not designed to handle language expression and cross-modal association with the above properties preserved.

**Vision and Language Pre-training (VLP).** Existing VLP works can be classified into two streams. One is based on the dual-encoder architecture, such as CLIP (Radford et al., 2021) and ALIGN (Jia et al., 2021), utilizing features encoded in each modality followed by contrastive learning (Oord et al., 2018) for alignment purposes. On the other hand, models like Zhou et al. (2020); Li et al. (2020); Zhang et al. (2021); Yang et al. (2021) exploit multiple cross-attention layers to learn the relationship between images and text, and show impressive performances on image-text matching tasks. Due to the efficiency of the former models in handling data across modalities, we adopt CLIP for cross-modal association in this paper.

## 6 CONCLUSION

A visual-linguistic learning framework with improve d adequacy, fidelity, and fluency (VLAF2) is presented for novel object captioning. We fundamentally quantify the above properties and relate them to the models of BERT and CLIP. We propose objectives and rewards reflecting the desirable linguistic fluency and visual semantics for NOC. Guided by BERT, our model learns to refine wordings of novel objects. Via reinforce algorithms, we have CLIP-based rewards assess the correctness of visual content described in the generated caption. Empirically, we showed that our model achieved SOTA results on the nocaps benchmark. We further provided analyses on both BERT and CLIP, verifying the necessity of their integration for learning novel object captions. A future direction of this work would be utilizing both captioned and uncaptioned images with self-supervised learning strategies for training NOC models.

## ETHICS STATEMENT

We acknowledge that all authors of this work have read and commit to adhering to the ICLR code of ethics. Our work has no ethical concern.

## REPRODUCIBILITY STATEMENT

For reproducing our results in the experiment section, codes and training/testing scripts are provided in the supplementary materials.

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

---

**Algorithm 1:** Learning to caption novel objects with linguistic fluency

---

**Input:** Captioning model $C_\theta(\cdot)$, Pre-trained BERT, and Pre-trained CLIP.
**Data:** Captioned image $x_l$, the corresponding GT caption $y_l$, uncaptioned image $x_u$, and lr $\eta_{it}$.
**Output:** Trained Captioning model $C_\theta(\cdot)$.

1   Initialize $C_\theta(\cdot)$;
2   **for** *it from 1 to num_iters* **do**
3      $\hat{y}_l \leftarrow C_\theta(x_l)$, $\hat{y}_u^c \leftarrow C_\theta(x_u)$;
4      Produce $\hat{y}_u^m$ by randomly masking words in the sentence in $\hat{y}_u^c$ (except for nouns);
5      $\hat{y}_u^b \leftarrow \text{BERT}(\hat{y}_u^m)$
6      $\mathcal{L}_{s2s} \leftarrow \text{CrossEntropy}(\hat{y}_l, y_l)$
7      **if** $CLIP(x_u, \hat{y}_u^b) \leq CLIP(x_u, \hat{y}_u^c)$ **then**
8         $\mathcal{L}_{BERT} \leftarrow 0$
9      **else**
10         $\mathcal{L}_{BERT} \leftarrow \text{CrossEntropy}(\hat{y}_u^c, \hat{y}_u^b)$
11      **end**
12      $\mathcal{L} \leftarrow \mathcal{L}_{s2s} + \mathcal{L}_{BERT}$
13      Update parameters: $\theta \leftarrow Adam(\theta, \eta_{it}, \nabla_\theta \mathcal{L})$
14   **end**

---

**Algorithm 2:** Learning novel object captions with fidelity and adequacy

---

**Input:** Captioning model $C_\theta(\cdot)$ and Pre-trained CLIP.
**Data:** Captioned image $x_l$, the corresponding GT caption $y_l$, uncaptioned image $x_u$, and lr $\eta_{it}$.
**Output:** Trained Captioning model $C_\theta(\cdot)$.

1   Initialize $C_\theta(\cdot)$;
2   **for** *it from 1 to num_iters* **do**
3      $\hat{y}_l^s \leftarrow C_\theta(x_l)$, $\hat{y}_u^s \leftarrow C_\theta(x_u)$ (by sampling);
4      $\hat{y}_l^g \leftarrow C_\theta(x_l)$, $\hat{y}_u^g \leftarrow C_\theta(x_u)$ (by greedy decoding);
5      Calculate $r_{\text{rep}}(\hat{y}_u^s)$ and $r_{rep}(\hat{y}_u^g)$ by (4)
6      $r(\hat{y}_l) \leftarrow r_{\text{CIDEr}}(\hat{y}_l, y_l) + r_{\text{CLIP}}(x_l, \hat{y}_l)$
7      $r(\hat{y}_u) \leftarrow r_{\text{CLIP}}(x_u, \hat{y}_u) + r_{\text{rep}}(\hat{y}_u)$
8      Calculate the gradient $\nabla_\theta \mathcal{L}_{RL}(\theta) \leftarrow -(r(\hat{y}_d^s) - r(\hat{y}_d^g))\nabla_\theta \log p_\theta(\hat{y}_d^s), d \in \{l, u\}$
9      Update parameters: $\theta \leftarrow Adam(\theta, \eta_{it}, \nabla_\theta \mathcal{L}_{RL})$
10   **end**

---

## A   IMPLEMENTATION DETAILS

Following Hu et al. (2020); Li et al. (2020); Zhang et al. (2021), we consider a BERT-base (Devlin et al., 2018) architecture for our captioning model. Given an image, the captioning model jointly takes the image region features and the predicted detection tags to generate the associated caption. We use the same region features as VinVL (Zhang et al., 2021), which are released on their project page. Since the object detection model Omni-detection used in previous works (Hu et al., 2020; Zhang et al., 2021) is not available, we replace it with a publicly available model of TSD (Song et al., 2020) to generate the object detection tag.

**Reproducing our method.** We perform VIVO (Hu et al., 2020) pre-training for 100 epochs with a batch size of 1024 and a learning rate of $5 \times 10^{-5}$, which are exactly the same as the parameters stated in the VIVO paper. After that, we propose to train our model following the training process described in Algorithm 1 to learn to caption novel objects with linguistic fluency. We train our model for 20 epochs with an effective batch size of 512 (256 caption-labeled images and 256 uncaptioned images) and a learning rate of $1.5 \times 10^{-5}$. Then, to learn novel object captions with fidelity and adequacy, we train our model as decsribed in Algorithm 2. Specifically, we train our model for 4 epochs with an effective batch size of 128 (64 caption-labeled images and 64 uncaptioned images) and a learning rate of $2.5 \times 10^{-6}$. We use 8 V100 GPUs to perform the above training algorithms. Codes can be found in the supplementary materials.

Table 5: Image captioning evaluation results on COCO "Karpathy" test split.

|  | BLEU@4 | METEOR | ROUGE-L | CIDEr | SPICE |
|---|---|---|---|---|---|
| VinVL | 39.8 | 29.9 | 59.6 | 134.6 | 23.9 |
| VinVL+VIVO | 39.7 | 29.9 | 59.6 | 134.5 | 23.8 |
| **Ours** | **40.0** | **30.4** | **60.2** | **137.3** | **24.5** |

Table 6: Ablation studies on nocaps validation set.

| Method | in-domain | | near-domain | | out-of-domain | | overall | |
|---|---|---|---|---|---|---|---|---|
|  | CIDEr | SPICE | CIDEr | SPICE | CIDEr | SPICE | CIDEr | SPICE |
| Baseline (Only w/ $L_{s2s}$) | 89.07 | 13.29 | 83.29 | 12.61 | 68.77 | 10.59 | 81.17 | 12.32 |
| + $L_{BERT}$ | 92.46 | 13.40 | 85.79 | 12.92 | 73.21 | 11.40 | 84.20 | 12.69 |
| + $r_{\text{CIDEr}}$ | 101.19 | 13.84 | 95.38 | 13.44 | 83.24 | 12.06 | 93.75 | 13.23 |
| + $r_{\text{CLIP}}$ | 96.73 | **14.83** | 89.64 | 14.12 | 81.87 | 12.38 | 89.08 | 13.88 |
| + $r_{\text{rep}}$ (Ours) | **102.77** | **14.83** | **97.90** | **14.40** | **86.33** | **12.54** | **96.25** | **14.10** |

**Reproducing baseline methods.** For VinVL (Zhang et al., 2021), we leverage the released model on their project page and directly inference on the nocaps dataset. However, for VinVL+VIVO (Zhang et al., 2021), since the pre-trained model is not publicly available, we reproduce this method using the image region features and object detection tags generated by models mentioned in the beginning of this section to train this model. Specifically, the model is trained for 160K iterations (about 100 epochs) with a batch size of 1024 and a learning rate of $5 \times 10^{-5}$, and fine-tuned for 30 epochs with a batch size of 256 and a learning rate of $5 \times 10^{-5}$ using the cross-entropy loss. Last, we perform the SCST optimization (Rennie et al., 2017) with a learning rate of $2 \times 10^{-6}$ for 5 epochs to obtain the final model. The numbers reported in Table 1 are derived using this version of model.

# B ADDITIONAL EXPERIMENTS

## B.1 EXPERIMENTS ON THE COCO CAPTION DATASET

To validate that our method generalize well on the task of describing the seen objects, we conduct experiments on the COCO Caption test set and report the numbers in Table 5. The training data for VinVL (Zhang et al., 2021) is image caption pairs from the COCO (Lin et al., 2014) dataset. While for VinVL + VIVO and our method, we additionally leverage the uncaptioned image from the Open Images (Kuznetsova et al., 2020) dataset as extra data. One can see that our method outperforms the other competitive approaches on different metrics which verifies the effectiveness of our approach.

## B.2 DETAILED ABLATION ANALYSIS

Table 6 lists the performances and compares contributions of the imposed objectives in our VLAF2. The baseline model in Table 6 is only trained on the COCO Caption dataset using the sequence-to-sequence objective. To confirm our introduction of exploiting BERT to learn the associated wordings of novel images, we apply $L_{BERT}$ to the baseline model, and report the results in the second row of Table 6. The CIDEr scores improve significantly after adopting reinforce algorithm (Williams, 1992) and using CIDEr scores of the generated captions as reward, and the results are in the third row. One can see that the SPICE scores largely increase but the CIDEr scores slightly decrease after the deployment of CLIP. We hypothesize that the captioning model properly captures the visual content in images, but it describes the scene with poor linguistic fluency. As the discussion in Sec. 3.2, we attribute the performance drop to the degenerate solution of increasing the association between the captions and the corresponding images. Note that we further consider the repetition penalty to regularize the captioning model. The results are shown in the last row of Table 6. One can see that this regularization slightly improves the SPICE scores but significantly increase the CIDEr scores. By comparing the performances listed in Table 6, we see that the full version of our VLAF2 achieved the best performance in terms of CIDEr and SPICE. Thus, the design of our VLAF2 can be successfully verified.

Table 7: Quantitative results on the nocaps (XD) test set.

| Method | overall | |
|---|---|---|
| | CIDEr | SPICE |
| UpDown (Agrawal et al., 2019) | 73.09 | 11.20 |
| SimVLM$_{base}$ (Wang et al., 2021b) | 94.80 | 13.10 |
| VIVO (Hu et al., 2020) | 100.12 | 14.04 |
| Ours | 96.25 | 14.10 |
| Ours (+CC) | **102.39** | **14.71** |

Table 8: Ablation studies of the joint-training model on nocaps validation set.

| Method | in-domain | | near-domain | | out-of-domain | | overall | |
|---|---|---|---|---|---|---|---|---|
| | CIDEr | SPICE | CIDEr | SPICE | CIDEr | SPICE | CIDEr | SPICE |
| Baseline (Only w/ $L_{s2s}$) | 96.1 | 13.71 | 90.35 | 13.41 | 79.96 | 11.77 | 89.07 | 13.13 |
| + $L_{BERT}$ | 99.44 | 13.91 | 91.13 | 13.53 | 81.11 | 11.82 | 90.29 | 13.25 |
| + $r_{CIDEr}$ | 109.14 | 14.52 | 100.66 | 14.08 | 88.61 | 12.69 | 99.43 | 13.87 |
| + $r_{CLIP}$ | 103.81 | **15.99** | 98.91 | **15.32** | 93.17 | **13.67** | 98.45 | **15.09** |
| + $r_{rep}$ (Ours) | **110.56** | 15.23 | **105.16** | 14.81 | **96.22** | 13.19 | **104.12** | 14.55 |

## B.3 EXPERIMENTS ON THE NOCAPS (XD) BENCHMARK

To investigate the limits of performance on nocaps without any restraints on the training datasets, we conduct experiments on the nocaps (XD) benchmark to verify the effectiveness of our method when more image-caption pairs are considered. Specifically, we additionally consider Conceptual Captions (CC) (Sharma et al., 2018) as labeled training samples $X_l$ in our learning framework and perform joint-training to see if extra image-caption pairs benefit the model on novel object captioning. The results are shown in Table 7. Note that both VIVO, SimVLM and our method adopt BERT-based architecture as the backbones for captioning. As for the training set, since VIVO (Hu et al., 2020) did not specify the dataset details for their evaluation for the nocaps (XD), we compared our method to SimVLM, which applied a much larger web-scale dataset (1.8B image-text pairs) than the CC dataset (3.1M pairs). Yet, our method still performs favorably against SOTAs on the nocaps (XD) protocol and benchmark, verifying the effectiveness of our method even if more image-caption pairs are considered.

In addition, to quantitatively show that the performance gain in Table 7 is not simply contributed by the additional data we considered, we ablate our model on the nocaps validation set and show the results in Table 8. We observed a similar performance trend as we reported in Table 6, where $L_{BERT}$ slightly improves the CIDEr scores, and $r_{CLIP}$ siginificantly boost SPICE but slightly deteriorates the CIDEr scores. One can see that the regularization $r_{rep}$ slightly improves the SPICE scores but significantly increase the CIDEr. By comparing the performances listed in Table 6 and Table 8, we see that our design of distilling BERT to enhance fluency (in terms of CIDEr) and the uses of CLIP to encourage captions with sufficient fidelity and adequacy (in terms of SPICE) still function properly when more diverse image-caption pairs are considered, verifying the design of our VLAF2.

## B.4 HUMAN STUDY

To conduct human study, we randomly picked 60 images from the nocaps validation set, and compared the captions generated by our method to those generated by the SOTA of VinVL+VIVO (Zhang et al., 2021), and the human-annotated captions provided by the nocaps dataset. Following the evaluation protocols used in the COCO Captioning Challenge 2015 (Lin et al., 2014), we designed 4 different metrics and asked individuals to evaluate captions from these aspects. The following are the four metrics we used in the experiment: M1: Is the caption generated by human (0: machine, 1: human)? (Percentage of captions that pass the Turing Test.) M2: Rate the correctness of the captions on a scale 1-5 (incorrect-correct): Whether the described objects or activities are correct. M3: Rate the amount of detail of the captions on a scale 1-5 (lack of details - very detailed): Whether the caption has detailed all the objects and their attributes. M4: Rate the fluency

Table 9: Human study on the nocaps validation set.

| Method | M1 (Turing Test) | M2 (Fluency) | M3 (Fidelity) | M4 (Adequacy) |
|---|---|---|---|---|
| VinVL+VIVO | 0.25 | 3.99 | 3.70 | 3.46 |
| Ours | 0.43 | 4.06 | 4.33 | **4.24** |
| Human | **0.53** | **4.09** | **4.44** | 4.18 |

of the captions on a scale 1-5 (lack of fluency-very fluent): Whether the caption use phrases/words that human generally would use to describe the scene, i.e., the caption is linguistically natural and fluent.

Specifically, M2, M3, M4 correspond to the fidelity, adequacy, and fluency, respectively, which are the particular objectives desired to be achieved. We asked 24 people two answer 6 different questionnaires, and each questionnaire contains 10 captions from each method (i.e., ours, sota, and human caption presented in a random order). We report the results in Table 9. We see that our method surpassed the SOTA by clear margins, while our performances were comparable to those the human ones across different metrics. This further supports the design of our model for NOC with sufficient fluency, fidelity, and adequacy.

### B.5 MORE QUALITATIVE RESULTS

**Qualitative comparison on fluency, fidelity and adequacy.** In this part, we provide more qualitative results on the nocaps validation/test set, and the results are shown in Fig. 5 and 6. Note that wordings that are less accurate or incorrectly describe the associated visual content are marked in bold. And, our wording improvements are highlighted in red. Take results in the bottom row of Fig. 5 for example. For the column of fluency, our model particularly described the turtle being *"crawling on some rocks"* instead of *"sitting on the top of a beach"*. For fidelity, our model predicted the background preferably as *"race track"* instead of *"street"* from the prediction of VinVL model. As for the column of adequacy, though both captions described a young men running, our model successfully captures more details in the image (i.e., *"there are number on their shirts"*). For more qualitative results, please refer to Fig. 6.

**Qualitative results of some failure cases.** In this part, we demonstrate some failure cases of our VLAF2 model. We empirically observe that the failure cases mainly come from the wrong/missing detection tags predicted by the pre-trained object detectors. To be more specific, the captioning model largely relies on the detection tags as clues to correctly describe novel objects. Take the result in the left-side of Fig. 7 for example, the detection model falsely recognizes the raccoon as a squirrel, and this detection result consequently damages the caption prediction. Therefore, how to jointly improve the detection model and captioning model is still a open question, and we leave this problem for future research. For more failure cases, please refer to Fig. 7.

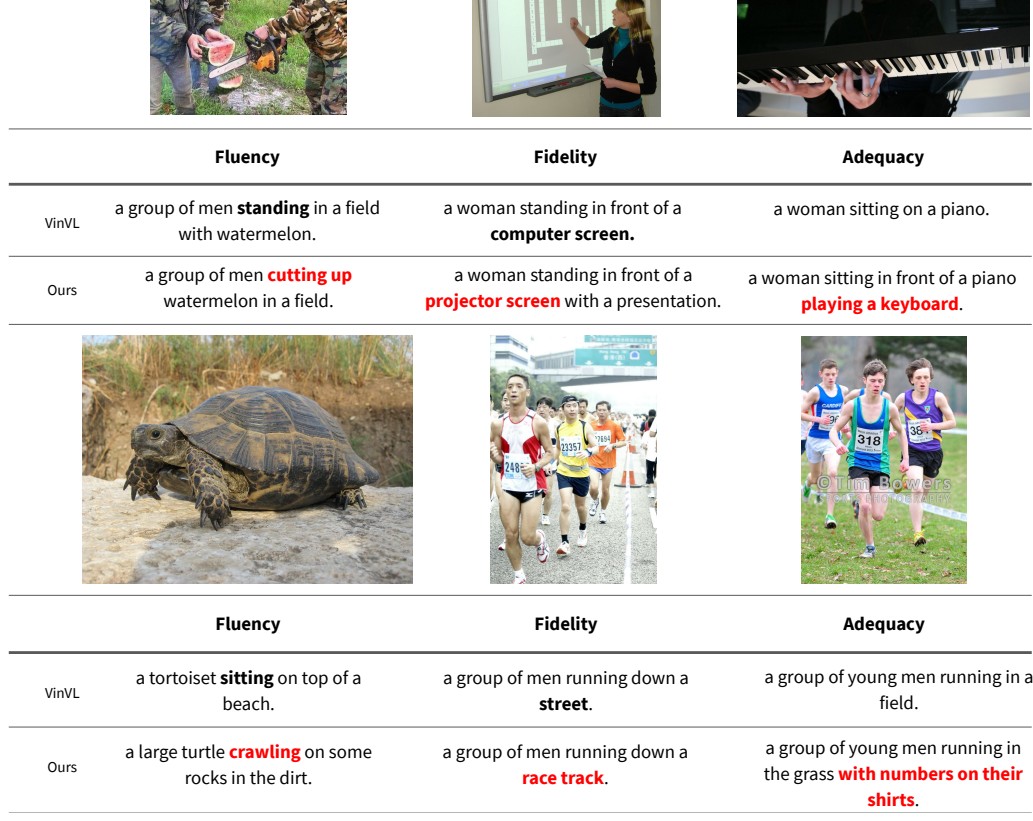

Figure 5: Example results and comparisons for image captions produced by VinVL and ours in terms of fluency, fidelity and adequacy. Note that both utilize VIVO for novel object detection.

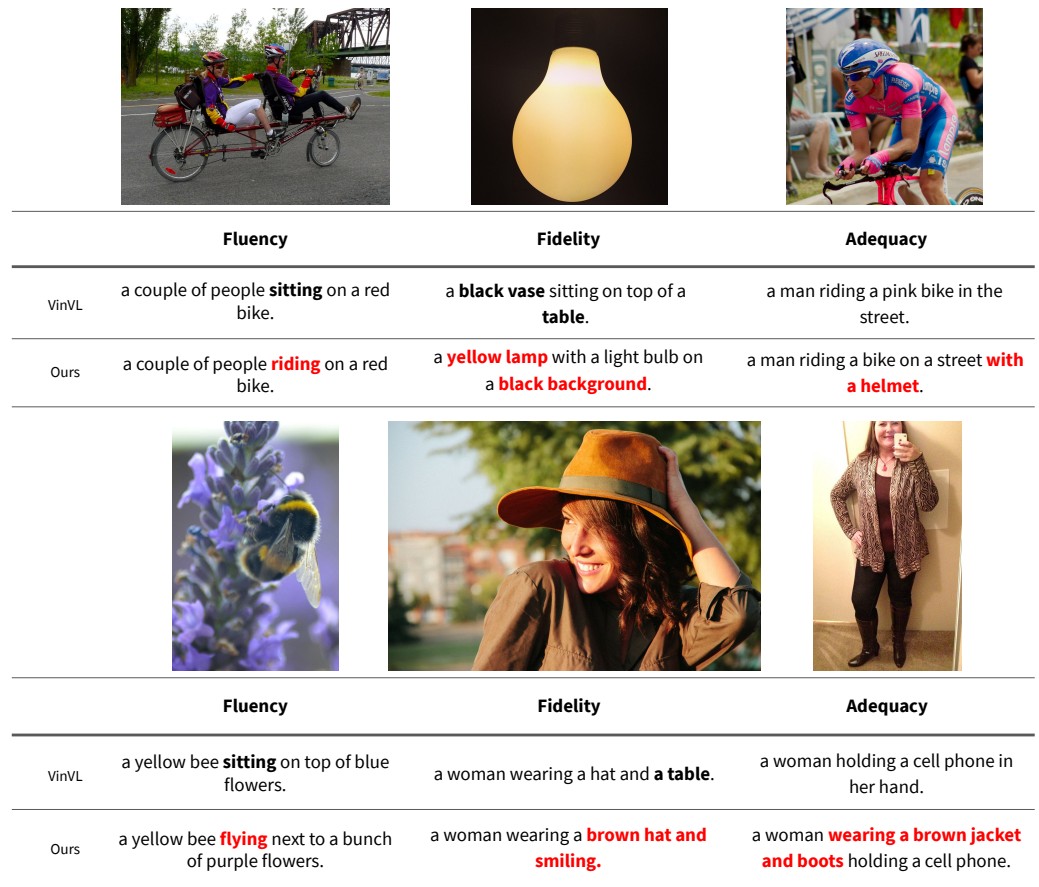

Figure 6: Example results and comparisons for image captions produced by VinVL and ours in terms of fluency, fidelity and adequacy. Note that both utilize VIVO for novel object detection.

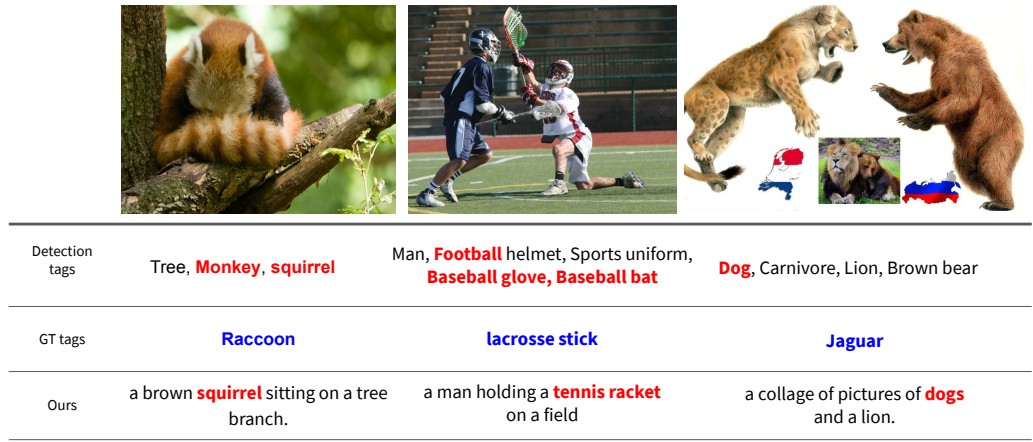

Figure 7: False captions misled by the wrong object detection tags.

