# OpenReview forum: "Learning Visual-Linguistic Adequacy, Fidelity, and Fluency for Novel Object Captioning"
_ICLR.cc/2022/Conference — ICLR 2022 Submitted_

### Official Review · Reviewer_1659 · 2021-10-27

**Correctness:** 3
**Technical Novelty And Significance:** 2
**Empirical Novelty And Significance:** 2
**Recommendation:** 5
**Confidence:** 4

**Main Review:**

1. Novel object captioning is one sub-task of image captioning and generating captions that are more fluency, fidelity, and adequacy is one sub-requirement of captioning. However, this method exploits two large-scale models BERT and CLIP to solve this small task, which is not an economic solution. In this way, I think the motivation of the research is not quite significant. Actually, not only novel object captions are not fluency, fidelity, and adequacy, even the captions generated by standard captioning settings (trained by MSCOCO) may also have low fluency, fidelity, and adequacy (Or say that standard captions' fluency, fidelity, and adequacy can be further improved). It would be better if the authors could apply their method to standard captioning or more settings which have limited training data other than only one novel object setting.
2. In Sec 3.1, the authors mention that they mask certain words instead of nouns since nouns may appear in the image, which should not be generated by BERT. However, the attribute and relation words may also appear in the image, which should not be masked. For example, some horses may be white and some others may be dark. Or in some images, a man ''ride'' a horse while in another image a man ''stand near'' a horse. Then these attributes and relations should also not be masked.
3. At the end of the first paragraph in Sec 3.2, the author claims that ''word-by-word supervision tend to imitate the sentence patterns of the training images instead of relating caption data to the visual content'' while do not provide suitable evidence.


**Summary Of The Paper:**

This paper tries to generate novel object captions that meantime satisfy three aspects: fluency, fidelity, and adequacy. To achieve this goal, it exploits two large-scale pertaining models which are BERT and CLIP. Specifically, given a novel object caption, the method first masks some words (expect nouns) in the sentence and then use BERT to generate these masked words. After that, they compare the association degree between two captions (the original caption and BERT re-generated caption) and the image. Such association degree is computed by inputting the image and the caption into CLIP. Based on the association degree, the method decides whether the BERT generated caption should be used as training supervision.

**Summary Of The Review:**

The major concern is about the significance of the motivation and the applied solution is not very economic by using two large scale pre-training models to solve a small task.

---

> ### Author Response · Authors · 2021-11-23
> **Response to Reviewer 1659 (Part 1 of 2)**
>
> We thank Reviewer 1659 for the suggestive and critical comments, which help us clarify and strengthen our work. Please see our response to each below.
>
> **Q1**: "Novel object captioning is one sub-task of image captioning, and generating captions that are more fluency, fidelity, and adequacy is one sub-requirement of captioning. I think the motivation of the research is not quite significant, but the applied solution is not very economic by using two large scale pre-training models to solve a small task. It would be better if the authors could apply their method to standard captioning or more settings which have limited training data other than only one novel object setting."
>
> **A1**: We thank the reviewer for bringing up this issue. We respectively disagree that NOC is a small task to address. NOC is actually a far more challenging image captioning task, addressed by researchers in recent years [5,6,7]. Nevertheless, we are more than happy to clarify the raised issues with quantitative supports.
>
> We agree that NOC can be viewed as a sub-task of general image captioning. However, solving a sub-task does not necessarily imply reduced difficulty when compared to that of solving the general task. For example, designing and training CNNs in a semi-supervised fashion would be more challenging than doing so in a fully-supervised setting. For NOC, how to produce image captions with fluency, fidelity, and adequacy guarantees *without* observing ground truth captions containing novel objects would be the major obstacles to tackle. Thus, we do want to first point out that NOC is a far more challenging task to be addressed when compared to general image captioning, as noted in recent works like [5,6,7].
>
> To realize the above objectives (i.e., enforcing caption fluency, fidelity and adequacy in NOC settings), we uniquely present fundamental supports in Sect. 2, explaining how we relate BERT and CLIP and thus leverage their intrinsic knowledge. The former is to enhance the fluency of the caption, while the latter is used to optimize for fidelity and adequacy (i.e., semantic correctness of the caption). It is worth noting that, using pre-trained models for deep learning problems is a widely accepted technique (e.g., image classification, question answering, etc.). Since our proposed learning scheme does *not* require updating BERT and CLIP, nor the requirement of auxiliary web-scale training data as [8] did, our method is practically not more computationally heavier than SOTAs for NOC.
>
> Finally, we would like to point out that, additional results on general image captioning (using COCO Captioning dataset) have been presented in Table 5 & 6 in Appendix B.1. We now present them here again for explanation and comparison purposes. Note that numbers before the slashes are results on the nocaps validation set (i.e., the task of NOC), and numbers after the slashes are results on the COCO Captioning “Karpathy” test split [9] (i.e. the task of general image captioning). The numbers in the second table were reported on the task of general image captioning (on COCO Captioning) in terms of fluency, fidelity and adequacy. From the results listed in the table below, we see that our method surpasses previous SOTAs on each of the captioning metrics in the below table, verifying the effectiveness of our method. Specifically, the improvements on CIDEr (first table) and fluency (second table) validates our claim of using BERT to enhance fluency, and the performance boost on SPICE (first table), fidelity, and adequacy (second table) demonstrates our design of using CLIP to improve fidelity and adequacy.
>
> |            |    BLEU@4     |    METEOR     |    ROUGE-L    |     CIDEr      |     SPICE     |
> | ---------- |:-------------:|:-------------:|:-------------:|:--------------:|:-------------:|
> | VinVL [1]  |   24.2/39.8   |   26.6/29.9   |   55.8/59.6   |   86.2/134.6   |   12.5/23.9   |
> | VinVL+VIVO |   26.3/39.7   |   27.8/29.9   |   56.1/59.6   |   89.3/134.5   |   12.6/23.8   |
> | Ours       | **26.6/40.0** | **28.5/30.4** | **57.0/60.2** | **96.3/137.3** | **14.1/24.5** |
>
> |            | Fluency  | Fidelity | Adequacy |
> | ---------- |:--------:|:--------:|:--------:|
> | VinVL      |   46.6   |   67.1   |   85.7   |
> | VinVL+VIVO |   45.8   |   65.6   |    86    |
> | Ours       | **47.6** |  **72**  | **86.9** |

---

> > ### Author Response · Authors · 2021-11-23
> > **Response to Reviewer 1659 (Part 2 of 2)**
> >
> > **Q2**: "In Sec 3.1, the authors mention that they mask certain words instead of nouns since nouns may appear in the image, which should not be generated by BERT. However, the attribute and relation words may also appear in the image, which should not be masked."
> >
> > **A2**: We thank the reviewer for pointing out this potential issue. In Eq. (2), BERT is utilized to randomly replace non-object words to improve the fluency of the caption, following a gating function by CLIP to verify whether such a replacement would be semantically consistent with the visual concepts of the associated image. This is because in NOC, not only the novel object labels (nouns) need to be discovered, producing their associated attributes or verbs are *equally important*. To ensure that the output caption would be linguistically correct (i.e., fluency). Since BERT is learned/pre-trained on web-scale datasets like the BookCorpus and the Wikipedia Corpus, we leverage its intrinsic linguistic knowledge to capture the co-occurrence of the object labels and their associated adjectives or verbs in the output caption. Again, while this allows us to preserve sufficient fluency, our model takes CLIP as guidance with verifies the consistency of the visual concepts described in the output caption.
> >
> > **Q3**: "At the end of the first paragraph in Sec 3.2, the author claims that *''word-by-word supervision tend to imitate the sentence patterns of the training images instead of relating caption data to the visual content''* while do not provide suitable evidence."
> >
> > **A3**: We thank the reviewer for giving us the opportunity to clarify this issue. In fact, this statement has been widely discussed in relevant literature [1,2,3,4] (i.e., standard sequence-to-sequence objectives for image captioning assign equal weights to each word in a caption, tending to produce outputs fitting generic sentence patterns instead of the desirable visual concept). In order to address this problem, we propose to adopt reinforce algorithms to optimize the captioning model could effectively solve this issue.
> >
> > For specifically, as described in Sect. 3.2, we introduce the objective in Eq. (5) which directly optimize the NOC model using the captioning metrics of CIDEr (r_CIDEr) and the cross-modal association (r_CLIP) as rewards. In our original manuscript, we have provided quantitative evidence in Table 6 and Appendix B to verify such model designs. We now highlight and summarize them below again for explanation purposes. In the following table, we consider the baseline model (i.e., our model without Eq. (5)), which observes labeled data with word-by-word supervision during training. From this table, we see that the deployment of our introduced reinforce algorithm remarkably benefited both CIDEr and SPICE scores. Compared to the baseline, our proposed model exhibits improved capability in producing captions containing desirable visual content for novel object images with sufficient linguistic fluency.
> >
> >
> > |                    | in domain |       | near domain |       | out domain |       | overall |       |
> > |--------------------|:---------:|:-----:|:-----------:|:-----:|:----------:|:-----:|:-------:|:-----:|
> > |                    |   CIDEr   | SPICE |    CIDEr    | SPICE |    CIDEr   | SPICE |  CIDEr  | SPICE |
> > | Baseline  |   92.46   | 13.40 |    85.79    | 12.92 |    73.21   | 11.40 |  84.20  | 12.69 |
> > | +REINFORCE   |   **102.77**  | **14.83** |     **97.90**    | **14.40** |    **86.33**   | **12.54** |  **96.25**  | **14.10** |
> >
> > [1] Rennie et al., "Self-critical sequence training for image captioning", CVPR 2017.
> >
> > [2] Liu et al., "Show, tell and discriminate: Image captioning by self-retrieval with partially labeled data", ECCV 2018.
> >
> > [3] Li et al., "Meta learning for image captioning", AAAI 2019.
> >
> > [4] Yang et al., "Fashion captioning: Towards generating accurate descriptions with semantic rewards", ECCV 2020.
> >
> > [5] Hu et al., "VIVO: Visual Vocabulary Pre-Training for Novel Object Captioning", AAAI 2021.
> >
> > [6] Zhang et al., "VinVL: Revisiting Visual Representations in Vision-Language Models", CVPR 2021.
> >
> > [7] Agrawal et al., "nocaps: novel object captioning at scale", ICCV 2019.
> >
> > [8] Wang et al., "SimVLM: Simple Visual Language Model Pretraining with Weak Supervision", Arxiv 2021.
> >
> > [9] https://github.com/karpathy/neuraltalk2

---

> > > ### Comment · Reviewer_1659 · 2021-11-30
> > > **Response to response**
> > >
> > > I do not think novel object captioning is a very important task and the applied approach is more like an extension of CLIP and BERT. Also, since during the pre-training of CLIP, maybe some novel objects have been seen by the whole framework. Then the novel object captioning becomes no-novel object captioning.

---

> > > > ### Author Response · Authors · 2021-11-30
> > > > **Follow-up**
> > > >
> > > > We thank the reviewer for providing additional feedback at the last minute, which allows us to understand what the major concerns remain.
> > > >
> > > > We are sorry that the reviewer still considers the task of NOC not important, while we provided recent SOTA NOC works of ICCV’19 [1], AAAI’21 [2] and CVPR’21 [3] as the supports, which are conducted by researchers at Microsoft and Facebook AI Research.
> > > >
> > > > We also understand that CLIP is pre-trained on large-scale image-caption pair data, which could potentially include images containing novel objects. As pointed out and acknowledged by Reviewer PRhm, CLIP is only trained to associate image-caption data at the instance but not word level. For fair comparison purposes (as suggested by Reviewer PRhm), we provide additional experiments on nocaps (XD), which is designed to investigate the limitation of NOC performance with auxiliary training datasets allowed. With Conceptual Captions (3.1M image-caption pairs) utilized to train our NOC model, we summarize the results on the nocaps test set below in the table:
> > > >
> > > > |                           | CIDEr  | SPICE |
> > > > | --------------------------|:------:|:-----:|
> > > > | UpDown [1]                | 73.09  | 11.20 |
> > > > | VIVO [2]                  | 100.12 | 14.04 |
> > > > | SimVLM_base [4] | 94.80  | 13.10 |
> > > > | Ours                      | **102.39** | **14.71** |
> > > >
> > > > Note that VIVO, SimVLM and our method all adopt BERT-based backbones for captioning. Although VIVO did not specify the details of datasets additionally applied, SimVLM specifically pointed out that a much larger web-scale dataset (**1.8B image-text pairs**) was utilized. From the above table, we see that our method still performs favorably against SOTAs on the suggested protocol and benchmark.
> > > >
> > > > We have uploaded our results on the nocaps (XD) benchmark, and here is the link to the leaderboard: https://eval.ai/web/challenges/challenge-page/464/leaderboard/1300. We have also updated the above results in Table 7 with discussions in Appendix B.3 of our revised manuscript.
> > > >
> > > > Nevertheless, we sincerely appreciate Reviewer 1659 for raising up the above practical issues. With prior responses and the above explanations, we hope the reviewer would assess our work fairly.
> > > >
> > > > [1] Agrawal et al., "nocaps: novel object captioning at scale", ICCV 2019.
> > > >
> > > > [2] Hu et al., "VIVO: Visual Vocabulary Pre-Training for Novel Object Captioning", AAAI 2021.
> > > >
> > > > [3] Zhang et al., "VinVL: Revisiting Visual Representations in Vision-Language Models", CVPR 2021.
> > > >
> > > > [4] Wang et al., "SimVLM: Simple Visual Language Model Pretraining with Weak Supervision", Arxiv 2021.

---

### Official Review · Reviewer_2cGW · 2021-10-28

**Correctness:** 3
**Technical Novelty And Significance:** 2
**Empirical Novelty And Significance:** 2
**Recommendation:** 6
**Confidence:** 3

**Main Review:**

### Pros:
- Leveraging prior visual-linguistic knowledge from pre-trained BERT and CLIP models for the task of novel object captioning. The association between fidelity, fluency and adequacy with the training tasks of BERT and CLIP presents a good motivation for the method adopted.

- The reward based loss functions designed for encouraging caption novelty and fluency and to discourage repetitive captions is interesting and novel.

- The authors design fidelity/fluency and adequacy quantitative evaluation to measure the efficacy of their approach compared to previous work and show improvements.

### Clarifications:
- Is the BERT model finetuned with the captions from the nocaps dataset in eq 2?  How is the CLIP condition satisfied in eq2, what happens if both the captions are unreliable ? Is there a threshold above which the CLIP scores are compared ?

- The BERT equations also could fulfil the fidelity objective because of the gating function. Is this observed?

- Are the improvements from Eq5. coming because of the r_CIDER term in the labelled caption data? Are the results in Table 1 compared with CIDER optimization for baselines?

- What is observed if the reward is not calculated over the labeled training samples X_l?

- How will  the model perform if Eq 2 is performed during the inference stage only from a pre-trained captioning model?

- In fig 1, it is not very clear where the second caption “two glove hands...” comes from, is one from greedy decoding and other from sampling  ?

- Are the novel words mostly objects/nouns?

### Cons:

- While the method is effective and simple, there are certain ablations which could have been performed qualitatively and quantitatively to draw where the improvements come from.

- The improvements in the method over the SOTA are via CIDER optimization on labeled examples in Eq 5. If the baselines are trained with CIDEr optimization, it might improve overall performance as well.

- The motivation of improving fluency or adequacy is probably not only related to novel image captioning but for image captioning in general. Could this method help to address those problems there?

- As stated in the paper, that previous methods produce grammatically incorrect captions for NOC, that does not seem to be the case (eg in fig 4). There are details missing from baseline captions but they are still fluent.


**Summary Of The Paper:**

The paper proposes to improve novel object captioning i.e. describing objects/contents that are not seen during training. Specifically, the authors propose to correct the captions generated from captioning models using pre-trained BERT and CLIP models by rewarding the captions with precise and rich visual content. They show qualitative and quantitative improvements on CiDER and SPICE scores compared to previous work and propose to measure the fluency, fidelity and adequacy of the generated captions for a more comprehensive evaluation of novel object captioning.

**Summary Of The Review:**

The paper is well written with detailed experimental results to support the claim. The paper proposes interesting ideas to improve NOC and is marginally novel but there are still some concerns as stated in the main review.

---

> ### Author Response · Authors · 2021-11-23
> **Response to Reviewer 2cGW (Part 1 of 3)**
>
> We thank Reviewer 2cGW for the suggestive and critical comments, which help us clarify and strengthen our work. Please see our response to each  below.
>
> **Q1**: "While the method is effective and simple, there are certain ablations which could have been performed qualitatively and quantitatively to draw where the improvements come from."
>
> **A1**: We thank the reviewer for giving us the opportunity to clarify this issue. In our original manuscript and appendix, we have presented ablation studies which verify the our model designs, including (1) distillation from BERT (L_BERT) to improve fluency, (2) discriminative reward from CLIP (r_CLIP) for fidelity and adequacy, and (3) the repetition penalty of the generated caption (r_rep) for avoiding repetitive words in the caption, as presented in Table 3 (in terms of image captioning metric) and Table 4 (in terms of fluency, fidelity and adequacy). From the above ablation studies in our original manuscript, we concluded that L_BERT majorly benefits fluency, while r_CLIP improves fidelity and adequacy of the caption, and r_rep prevents the generated captions into a bag of object category labels.
>
> To make the above assessment more complete, we conducted additional ablation studies and presented in Appendix B.2 in our original manuscript (and as listed below). Instead of removing a single component from our model for ablation study (as our main paper does), we incrementally added each objective and reward to our learning framework, demonstrating the contribution of each component. From the table below, we observe the same trends that L_BERT benefits CIDEr, reflecting caption fluency by measuring the n-gram overlapping between prediction and ground-truth captions. The significant improvements on CIDEr validate its use as reward (r_CIDEr). In addition, r_CLIP encourages fidelity and adequacy of the captions, boosting the SPICE score which evaluates the visual content presented in the captions. Nevertheless, the full version of our model would achieve the highest performance across different domains and evaluation metrics. Thus, its model design, objectives, and learning strategies can be successfully verified.
>
> With existing and the additional ablation studies verifying the design of our proposed model and each of the introduced objectives, we would appreciate if the reviewers could be more specific on what additional ablation tables would be desirable.
>
> |                    | in domain |       | near domain |       | out domain |       | overall |       |
> |--------------------|:---------:|:-----:|:-----------:|:-----:|:----------:|:-----:|:-------:|:-----:|
> |                    |   CIDEr   | SPICE |    CIDEr    | SPICE |    CIDEr   | SPICE |  CIDEr  | SPICE |
> | Baseline            |   89.07   | 13.29 |    83.29    | 12.61 |    68.77   | 10.59 |  81.17  | 12.32 |
> | +L_BERT  |   92.46   | 13.40 |    85.79    | 12.92 |    73.21   | 11.40 |  84.20  | 12.69 |
> | +r_CIDEr |   101.19  | 13.84 |    95.38    | 13.44 |    83.24   | 12.06 |  93.75  | 13.23 |
> | +r_CLIP  |   96.73   | 14.83 |    89.64    | 14.12 |    81.87   | 12.38 |  89.08  | 13.88 |
> | +r_rep   |   **102.77**  | **14.83** |     **97.90**    | **14.40** |    **86.33**   | **12.54** |  **96.25**  | **14.10** |
>
>
>
> **Q2**: "Are the improvements from Eq. 5 coming because of the r_CIDER term in the labelled caption data? "
>
> **A2**: We thank the reviewers for giving us the opportunity to clarify this issue. The improvements from Eq. (5) is not simply contributed by the r_CIDEr term, as we explain below.
>
> Eq. (5) is calculated for both labeled and unlabeled caption data during training. For labeled data, the CIDEr reward r_CIDEr is observed to improve the caption *fluency*, while the CLIP reward r_CLIP is to exploit the visual content in the caption, which thus improves the corresponding fidelity and adequacy. For unlabeled caption data, we are only able to calculate r_CLIP not the CIDEr reward (r_CIDEr), because r_CIDEr cannot be calculated without ground-truth captions. Instead, we additionally enforce the repetition penalty r_rep to prevent model from repeating the same objects that occur in the image, which would simply increase r_CLIP as stated in Sect. 3.2 with examples. With existing and suggested ablation studies presented in **Q1**, we confirm that r_CIDEr contributes to the improvement in CIDEr, while the design of r_CLIP enhances the SPICE score by a significant margin. Finally, r_rep allows our model to produce semantically fluent captions, which thus further increases performance in terms of both CIDEr and SPICE.

---

> > ### Author Response · Authors · 2021-11-23
> > **Response to Reviewer 2cGW (Part 2 of 3)**
> >
> > **Q3**: "Are the results in Table 1 compared with CIDER optimization for baselines? The improvements in the method over the SOTA are via CIDER optimization on labeled examples in Eq 5. If the baselines are trained with CIDEr optimization, it might improve overall performance as well."
> >
> > **A3**: We thank the reviewer for pointing this out. In fact, the improvements of our method are not due to the use of CIDEr optimization. In Table 1 of the manuscript, we evaluate all methods (including SOTAs and ours) on the nocaps validation set, with all models trained via CIDEr optimization (i.e., Oscar [1], VIVO [2], VinVL[3], except the UpDown baseline [4] since they did not report such results). Thus, our improvements over the SOTA are not simply contributed by the CIDEr optimization.
> >
> > **Q4**: "The motivation of improving fluency or adequacy is probably not only related to novel image captioning but for image captioning in general. Could this method help to address those problems there?"
> >
> > **A4**: Yes. We agree with the reviewer that, the goal of improving caption fluency and adequacy would be shared by general image captioning tasks. Since our work focuses on NOC, we did conduct experiments on general image captioning dataset like COCO Captioning and reported the numbers originally in Table 5 in our Appendix. As suggested, we conduct such experiments and present the results here. Note that numbers before the slashes are results on the nocaps validation set (i.e., the task of NOC), and numbers after the slashes are results on the COCO Captioning “Karpathy” test split [5] (i.e., general image captioning). In the task of general image captioning, distillation from BERT would serve as auxiliary guidance to enhance caption fluency, and our CLIP reward encourages models to produce descriptive and discriminative captions. From the table below, one can see that our method again performs favorably against SOTAs on each captioning metric, which confirms the generalization of our model design and learning strategies for general image captioning.
> >
> > |            |    BLEU@4     |    METEOR     |    ROUGE-L    |     CIDEr      |     SPICE     |
> > | ---------- |:-------------:|:-------------:|:-------------:|:--------------:|:-------------:|
> > | VinVL [3]  |   24.2/39.8   |   26.6/29.9   |   55.8/59.6   |   86.2/134.6   |   12.5/23.9   |
> > | VinVL+VIVO [2,3] |   26.3/39.7   |   27.8/29.9   |   56.1/59.6   |   89.3/134.5   |   12.6/23.8   |
> > | Ours       | **26.6/40.0** | **28.5/30.4** | **57.0/60.2** | **96.3/137.3** | **14.1/24.5** |
> >
> > To further address the concerns of the reviewer, we additionally evaluate our method on the task of general image captioning (on COCO Captioning) in terms of fluency, fidelity and adequacy (with the results presented in the following table). Thus, from the above existing and additional experiments, it can be verified that our model design of distilling BERT (to preserve fluency) and using CLIP for rewarding captions (to improve fidelity and adequacy) would also benefit the task of general image captioning.
> >
> > |            | Fluency  | Fidelity | Adequacy |
> > | ---------- |:--------:|:--------:|:--------:|
> > | VinVL      |   46.6   |   67.1   |   85.7   |
> > | VinVL+VIVO |   45.8   |   65.6   |    86    |
> > | Ours       | **47.6** |  **72**  | **86.9** |
> >
> > **Q5**: "As stated in the paper, that previous methods produce grammatically incorrect captions for NOC, that does not seem to be the case (eg in fig 4). There are details missing from baseline captions but they are still fluent."
> >
> > **A5**: We apologize for the confusion. Throughout our paper (e.g., Sections 1 and 2), we emphasize that a desirable NOC model should properly describe the visual concept of images containing object, with linguistically natural and grammatically correct (i.e., fluency). We did not imply grammatical incorrectness would be the major issue of SOTAs. As discussed in Sec. 4.2, methods like VinVL+VIVO did not produce satisfactory performance in terms of CIDEr/fluency (see Tables 3 & 4 as we did. And, Fig. 4 in our manuscript is used to demonstrate that our method is able to describe all desirable visual concepts with sufficient fluency when comparing the SOTA of VinVL (e.g., *a fan hanging from the ceiling in a room* (ours) vs. *a fan in a room* (VinVL+VIVO)).

---

> > > ### Author Response · Authors · 2021-11-23
> > > **Response to Reviewer 2cGW (Part 3 of 3)**
> > >
> > > **Q7**: "What is observed if the reward is not calculated over the labeled training samples X_l?"
> > >
> > > **A7**: We are happy to provide additional explanations and results to address this issue. We follow the suggestions from reviewer and conduct experiments without using labeled training samples X_l and we summarize the results here. We found that the suggested model would achieve degraded performance in terms of CIDEr and thus cannot achieve comparable performance as our full model did. We believe the reason is that, models disregarding rewards calculated for labeled data would lack the linguistic property of the caption. In other words, while focusing on maximizing the CLIP reward for boosting caption fidelity and adequacy, such model tend to produce captions with insufficient fluency and thus would not be desirable.
> > >
> > > Recall that in Eq. (5), the labeled training samples are rewarded with the CIDEr reward r_CIDEr and the CLIP reward r_CLIP, and the unlabeled data are rewarded with the CLIP reward r_CLIP and the repetition penalty r_rep. Without calculating the CIDEr reward for labeled data, model would easily increase the r_CLIP by repeating visual content in the caption, leading to degenerated linguistic fluency. Such tendency can be observed from CIDEr and SPICE scores listed in the following table. One can see that the performance drop on CIDEr is significant when model is optimized without labeled samples X_l since CIDEr compares predicted captions with the entire ground-truth sentences, considering the linguistic fluency of the caption. On the other hand, the SPICE score focuses on whether the visual content is preserved by the predicted captions and does not consider the linguistic fluency of the caption. Therefore, the slight improvement on SPICE verifies our claim of over-fitting on r_CLIP when training without labeled samples X_l. Thus, to achieve high CIDEr and SPICE at the same time, labeled samples X_l are indispensable for our learning framework.
> > >
> > > |             | CIDEr | SPICE |
> > > | ----------- |:-----:|:-----:|
> > > | Ours        | 96.3  | 14.1  |
> > > | Without X_l | 39.7  | 14.9  |
> > >
> > > **Q8**: "How will the model perform if Eq. 2 is performed during the inference stage only from a pre-trained captioning model?"
> > >
> > > **A8**: We are happy to carry out additional experiments as suggested. Recall that, Eq. (2) utilizes BERT to randomly replace a non-object word in the caption, with results verified by CLIP to ensure the replacement sufficiently describe the visual content. As suggested, we take the baseline model of VinVL [3] to perform this experiment. From the results shown in the table below, we observe that such replacements (performed during inference) only slightly increased caption fluency (in terms of CIDEr) but with negligible improvements on fidelity and adequacy (in terms of SPICE). Similar observations can be seen in Tables 3 & 6 of our manuscript.
> > >
> > > |                    | CIDEr | SPICE |
> > > | ------------------ |:-----:|:-----:|
> > > | Before replacement | 83.31 | 12.71 |
> > > | After replacement  | 83.92 | 12.72 |
> > >
> > > **Q9**: "In fig 1, it is not very clear where the second caption “two glove hands...” comes from, is one from greedy decoding and other from sampling ?"
> > >
> > > **A9**: Yes. As we depicted in Fig. 3, the first caption, “a blue shirt...” comes from greedy decoding, while the second caption, “two glove hands...” comes from sampling. We thank the reviewer for the suggestion, and we have modified Fig. 1 in our revised manuscript.
> > >
> > > **Q10**: "Are the novel words mostly objects/nouns?"
> > >
> > > **A10**: Yes, novel objects are nouns. However, not only predicting novel object labels, a desirable NOC model requires to predict the associated verbs or attributes, so that the produced captions would exhibit sufficient fluency, fidelity, and adequacy. These properties are the goals of our work and have been extensively & quantitatively verified in our manuscript.
> > >
> > >
> > > [1] Li et al., "Oscar: Object-Semantics Aligned Pre-training for Vision-Language Tasks", ECCV 2020.
> > >
> > > [2] Hu et al., "VIVO: Visual Vocabulary Pre-Training for Novel Object Captioning", AAAI 2021.
> > >
> > > [3] Zhang et al., "VinVL: Revisiting Visual Representations in Vision-Language Models", CVPR 2021.
> > >
> > > [4] Agrawal et al., "nocaps: novel object captioning at scale", ICCV 2019.
> > >
> > > [5] https://github.com/karpathy/neuraltalk2
> > >
> > > [6] Zhu et al., "Aligning Books and Movies: Towards Story-like Visual Explanations by Watching Movies and Reading Books", ICCV 2015.
> > >
> > > [7] https://www.english-corpora.org/wiki/
> > >
> > > [8] Xie et al., "Self-training with Noisy Student improves ImageNet classification", CVPR 2020.
> > >
> > > [9] Jia et al., "Scaling Up Visual and Vision-Language Representation Learning With Noisy Text Supervision", ICML 2021.
> > >
> > > [10] Chen et al., "Distilling Knowledge Learned in BERT for Text Generation", ACL 2020.

---

> ### Comment · Reviewer_2cGW · 2021-11-26
> **Response to Author Feedback**
>
> Thanks to the authors for the feedback and additional experiments, really appreciated. Most of my concerns are resolved after the new sets of experimental results, discussions and responses to the main questions raised.
> I have updated my final rating accordingly. Overall, the paper presents an interesting approach to improve the three main problems in image captioning and achieves improvement over stat-of-the-art. A small concern about novelty and reliance on heavy models (BERT and CLIP) still remains but an insight into the method using sufficent ablation studies and discussions convinces me to push the initial score.

---

> > ### Author Response · Authors · 2021-11-27
> > **Follow-up**
> >
> > Thank you for the positive feedback. We'd be happy to clarify if any further concerns raised.

---

### Official Review · Reviewer_F11g · 2021-11-01

**Correctness:** 3
**Technical Novelty And Significance:** 2
**Empirical Novelty And Significance:** Not applicable
**Recommendation:** 6
**Confidence:** 4

**Main Review:**

Strengths:

S1: This paper pays attention to three important properties of captions -- fluency, fidelity and adequacy and proposes a formwork to generate fluent, faithful and adequate captions to describe images.

S2: Extensive experimental results show that the proposed model outperforms existing models on novel object captioning.

Weaknesses:

W1: The proposed model combines some existing models, like BERT, CLIP, so one of my concerns is the novelty of the proposed framwork. Plus, the reward to train the model is not novel, which is used in Rotian's work [1] and distilling from BERT for text generation was proposed by Chen [2]. Also, the observation that repeating the objects that occur in an image can improve retrieval score is not new [3].

W2: Neigher BERT nor CLIP is trained using the dataset used in the paper, so how to mitigate the problem of domain shift when applying BERT and CLIP? I do not think you can assume that BERT and CLIP are always correct.

W3: It seems that BERT is only used during training. I am wondering whether BERT can be used in the inference stage, for example, using the captioning model to generate a caption with beam search and then employing BERT to generate more captions. In this way, we can see whether BERT can improve fluency.

W4: Lack of human study.

[1] Rotian Luo et al.. Discriminability objective for training descriptive captions. CVPR, 2018.

[2] Y-C Chen et al.. Distilling Knowledge Learned in BERT for Text Generation. ACL, 2020.

[3] Qingzhong Wang et al.. Describing Like Humans: on Diversity in Image Captioning. CVPR, 2019.

**Summary Of The Paper:**

This paper proposes a framework that combines masked language models (BERT) and image-text embedding models (CLIP) for novel object captioning. The proposed model significantly improves the fluency, fidelity, and adequacy of the generated images. Extensive experiments are conducted and the comparison between the proposed model and existing models shows that the proposed model performs relatively well.

**Summary Of The Review:**

This paper proposed a strong baseline of novel object captioning via combining BERT and CLIP models, however, the novelty is limited.

---

> ### Author Response · Authors · 2021-11-23
> **Response to Reviewer F11g (Part 1 of 3)**
>
> We thank Reviewer F11g for the positive comments and suggestive remarks. Please see our responses below for each raised issue.
>
> **Q1**: "The proposed model combines some existing models, like BERT, CLIP, so one of my concerns is the novelty of the proposed framework. Plus, the reward to train the model is not novel, which is used in Rotian's work and distilling from BERT for text generation was proposed by Chen. Also, the observation that repeating the objects that occur in an image can improve retrieval score is not new."
>
> **A1**: We understand the particular concern of novelty from the reviewer, and we are more than happy to clarify this issue. We would like to first point out that, we did not claim the uses of BERT, CLIP, or reward function designs being the contributions. We are well aware that such well-developed models have been applied for solving tasks like image-text retrieval (e.g., CLIP) or text generation (e.g., BERT).
>
> As noted in Sect. 1., we address the challenging task of novel object captioning (NOC), which requires captioning models to accurately describe images containing novel objects with captions related to such objects unseen during training. In our work, we first present fundamental supports in Sect. 2, explaining how we are able to utilize and relate BERT and CLIP for NOC, leveraging their intrinsic knowledge for producing captions with fluency, fidelity, and adequacy guarantees. With the designed objectives and learning strategies proposed in Sect. 3, we are able to quantitatively verify and support our model design (i.e., BERT can be used to optimize for caption fluency, while CLIP is beneficial for boosting both fidelity and adequacy in NOC).
>
> As for the reward function utilized, our introduced CLIP reward r_CLIP observes all possible image-caption pairs in a mini-batch, allowing our model to produce descriptive and discriminative captions. Rotian's work [5], as suggested by the reviewer, applies the standard triplet loss, which only considers one single positive and one single negative image-caption pair from the same mini-batch. We take the suggestion from the reviewer and replace r_CLIP in Eq. (5) with the triplet loss in [5], and present the comparisons in the first row of the following table. Compared to our model with SPICE of 14.10, the suggested reward of [5] only achieved 13.25, implying predicted captions with reduced discrimination ability. It suggests that the use of [5] for aligning image and caption data would not be desirable.
>
> As for the distillation technique proposed by Chen [6], we would like to note that it only works in single-modality scenarios (e.g., machine translation). In other words, it lacks the ability to perform distillation for tasks associated with cross-modal data (e.g., image and caption), and might result in caption data not reflecting the visual concept presented in the corresponding image. Again, we follow the suggestion by the reviewer and conduct additional experiments to assess the performance using the distillation technique of [6]. With the results listed and compared in the second row of the table, we again find the design of our proposed model to be preferable.
>
> Finally, as discussed in Sect. 3.2, we bring up the practical issue that captions which repeatingly describe particular objects would produce high image-text retrieval scores, and thus fail to reflect the linguistic correctness of the output caption. While we understand that this is an undesirable issue, we did *not* claim that such an observation is new. Instead, we propose to tackle this problem by imposing a repetition penalty r_rep, as defined in Eq. (4). With such additional regularization, promising NOC performance with improved CIDEr and SPICE scores can be observed in Table 3 (and the third row in the table below).
>
> |                                            |   CIDEr   |   SPICE   |
> | ------------------------------------------ |:---------:|:---------:|
> | Replace r_CLIP with triplet loss |   84.58   |   13.25   |
> | Eq. 2 w/o gated by CLIP                    |   21.43   |   9.43    |
> | w/o r_rep                     |   89.05   |   13.88   |
> | Ours                                       | **96.25** | **14.10** |

---

> > ### Author Response · Authors · 2021-11-23
> > **Response to Reviewer F11g (Part 2 of 3)**
> >
> > **Q2**: "Neither BERT nor CLIP is trained using the dataset used in the paper, so how to mitigate the problem of domain shift when applying BERT and CLIP? I do not think you can assume that BERT and CLIP are always correct."
> >
> > **A2**: We thank the reviewer for pointing out the potential issue of domain shift. We note that, both BERT and CLIP are pre-trained on *web-scale* datasets, describing general text or image-text data. The NOC task considered in [1,3] and our paper focuses on captioning image data from Open Images, which also contains general image data and aligns well with the uses of the aforementioned web-scale data for (pre-)training. Thus, compared to the description of specific image domains like medical or satellite data, domain shift would not be viewed as a particular concern by the above NOC works including ours (as we quantitatively verified in our experiments).
> >
> > To further address reviewer's concern, we additionally compare our model with [4] which applied much larger datasets (e.g., 1.8 billion in the dataset proposed by [2]) for training. With the results shown in the table below, we see that our method still achieved comparable performances on the nocaps test set in terms of both CIDEr and SPICE . Thus, the uses of BERT and CLIP for general NOC tasks would be sufficiently preferable and applicable.
> >
> > |                           |   CIDEr   |   SPICE   |
> > | ------------------------- |:---------:|:---------:|
> > | SimVLM_base [4] | **94.80** |   13.10   |
> > | Ours                      |   93.50   | **14.10** |
> >
> >
> > **Q3**: "It seems that BERT is only used during training. I am wondering whether BERT can be used in the inference stage. In this way, we can see whether BERT can improve fluency."
> >
> > **A3**: We thank the reviewer for the suggestion. As described in Sec. 3.1, BERT is utilized in our model, so that its linguistic knowledge would be distilled for preserving caption fluency. More precisely, it is utilized to replace non-object words in the generated, guided by the associated CLIP module for enforcing the generated caption to be sufficiently fluent. If BERT is further applied to repeat the same procedure (i.e., replacing non-object words) during inference, we do not expect to observe further improvements in caption fluency. Nevertheless, we followed the suggestion from reviewer and conducted experiments on the nocaps validation set. As shown in the table below, both the CIDEr and SPICE scores are with negligible changes, verifying that our model has properly leveraged the desirable linguistic knowledge during training.
> >
> > |                               | CIDEr | SPICE |
> > | ----------------------------- |:-----:|:-----:|
> > | Ours                          | 96.25 | 14.10 |
> > | Ours with replacement by BERT | 96.15 | 14.08 |

---

> > > ### Author Response · Authors · 2021-11-23
> > > **Response to Reviewer F11g (Part 3 of 3)**
> > >
> > > **Q4**: Lack of human study.
> > >
> > > **A4**: We thank the reviewer for giving us the opportunity to further improve our paper. As suggested, we conduct a series of human study to further verify the effectiveness and practicality of our proposed model for NOC.
> > >
> > > To conduct such experiments, we randomly picked 60 images from the nocaps validation set, and compared the captions generated by our method to those generated by the SOTA of VinVL+VIVO [4], and the human-annotated captions provided by the nocaps dataset. Following the evaluation protocol used in the COCO Captioning Challenge 2015: [link](https://cocodataset.org/#captions-leaderboard), we designed 4 different metrics and asked individuals to evaluate captions from these aspects:
> > >
> > > **M1**: Is the caption generated by human (0: machine, 1: human)? (Percentage of captions that pass the Turing Test.)
> > >
> > > **M2**: Rate the correctness of the captions on a scale 1-5 (incorrect-correct): Whether the described objects or activities are correct.
> > >
> > > **M3**: Rate the amount of detail of the captions on a scale 1-5 (lack of details - very detailed): Whether the caption has detailed all the objects and their attributes.
> > >
> > > **M4**: Rate the fluency of the captions on a scale 1-5 (lack of fluency -very fluent): Whether the caption use phrases/words that human generally would use to describe the scene, i.e., the caption is linguistically natural and fluent.
> > >
> > > Specifically, M2, M3, M4 correspond to the fidelity, adequacy, and fluency, respectively, which are the particular objectives desired to be achieved. We asked 24 people to answer 6 different questionnaires, and each questionnaire contains 10 captions from each method (i.e., ours, sota, and human caption presented in random order). We report the results below.
> > >
> > > |            | M1 (Turing Test) | M2 (Fidelity) | M3 (Adequacy) | M4 (Fluency) |
> > > | ---------- |:----------------:|:-------------:|:-------------:|:------------:|
> > > | VinVL+VIVO |       0.25       |     3.70      |     3.46      |     3.99     |=
> > > | Ours       |       0.43       |     4.33      |   **4.24**    |     4.06     |
> > > | Human      |     **0.53**     |   **4.44**    |     4.18      |   **4.09**  |
> > >
> > > From the above table, we see that our method surpassed the SOTA by clear margins, while our performances were comparable to those of the human ones across different metrics. This further supports the design of our model for NOC with sufficient fluency, fidelity, and adequacy.
> > >
> > > We have updated the above results in Table 9. and discussed in Appendix B.4 in our revised manuscript.
> > >
> > >
> > > [1] Hu et al., "VIVO: Visual Vocabulary Pre-Training for Novel Object Captioning", AAAI 2021.
> > >
> > > [2] Jia et al., "Scaling Up Visual and Vision-Language Representation Learning With Noisy Text Supervision", ICML 2021.
> > >
> > > [3] Zhang et al., "VinVL: Revisiting Visual Representations in Vision-Language Models", CVPR 2021.
> > >
> > > [4] Wang et al., "SimVLM: Simple Visual Language Model Pretraining with Weak Supervision", Arxiv 2021.
> > >
> > > [5] Luo et al., "Discriminability objective for training descriptive captions", CVPR 2018.
> > >
> > > [6] Chen et al., "Distilling Knowledge Learned in BERT for Text Generation", ACL 2020.

---

### Official Review · Reviewer_PRhm · 2021-11-01

**Correctness:** 3
**Technical Novelty And Significance:** 3
**Empirical Novelty And Significance:** 4
**Recommendation:** 6
**Confidence:** 5

**Main Review:**

### Strengths

1. Overall, I concur with the key insight from the paper which is that models for NOC should tackle fluency, fidelity and adequacy. It has been observed in Agrawal et al 2019 (nocaps dataset) that methods suffer from these issues. Captions which are fluent often aren't accurate, and captions which are accurate often aren't fluent.
2. The authors take reasonable steps to fix these issues. Using a LM to fix fluency is a good idea. The authors address one of the main issues with such an approach — BERT like language models can produce fluent captions but it's hard to judge their accuracy. The authors use CLIP to score these and select the best method. Such an approach makes sense.
3. Another novel contribution in this work is using uncaptioned images (images in the OpenImages dataset which do not have associated training captions) during training. Since these don't have training captions, they modify the reward to use CLIP as well penalize repeated words. Indeed, they show that CLIP + repeated penalty is helpful for overall performance (Table 3).

### Weaknesses

1. One of the biggest and most important weakness in the paper is that the paper doesn't follow the guidelines from the nocaps paper. As mentioned in the nocaps paper (section 3.3), "Do not use additional paired image-caption data collected from humans. Improving evaluation scores by leveraging additional human-generated paired imagecaption data is antithetical to this benchmark – the only paired image-caption dataset that should be used is the COCO Captions 2017 training split. However, external text corpora, knowledge bases, and object detection datasets may be used during training or inference". The paper heavily relies on CLIP which has been trained on alt-text data for images (which are human written) for improving the captioning performance. This makes the method proposed in the paper not comparable to existing methods in my opinion. The authors should have used the nocaps (XD) for evaluation.
2. Related to the second point, I would have liked to see if you can replace CLIP with other approaches (ViLBERT) which also outputs a score indicating how well the caption is aligned with the image. This would form a fair comparison.
3. Since CLIP is trained on alt-text data for images, it would be interesting to see if joint-training on Conceptual Captions (or a similar web-scale data) works better instead.
4. CLIP is being used for checking the relevance of the caption with the image. I wonder how will other similar metrics for caption relevance (VIFIDEL) compare against scores from CLIP.

**Summary Of The Paper:**

The papers presents an approach for the task of novel object captioning (NOC). NOC requires the model to describe novel objects appearing in test images that were not described in the training captions. The authors claim that existing methods are not good at the task because they don't tackle three concepts required to produce a good caption —

1) Fluency: generating human-like captions,

2) Fidelity: whether the caption is describing the novel object in the image or not,

3) Adequacy: whether the caption covers all the salient visual concepts in the image.

The authors propose a model Visual-Linguistic Adequacy, Fidelity, and Adequacy (VLAF2) to address these issues. The model utilizes BERT to sample different variations of the captions by substituting random words from the caption. It uses CLIP to assess which of these are more relevant to the image. The model also uses CLIP score and combine it with CIDEr based reward to further train the captioning model via reinforce.

The model is evaluated on the nocaps benchmark on which it achieves improvements over existing sota methods like VINVL+VIVO. The authors also show that apart from captioning metrics, the model also does better on fluency (B@4, C), Fidelity (Precision), Adequacy(Recall).

**Summary Of The Review:**

Overall, I think the paper presents an interesting insight (breaking down caption quality into fluency, adequacy and fidelity) and presents an approach which improves all the three aspects on the nocaps dataset. I raised a couple of concerns (use of CLIP which has been trained on human-generated alt-text data) which goes against the guidelines of the benchmark. It nullifies some of the claims in the paper and I'd be interested in hearing from the authors regarding that. I would have also liked to see comparison with other caption relevance metrics like VIFIDEL which would make the contributions stronger. Overall, I am borderline on this paper and will update my thoughts based on the author response.

---

> ### Author Response · Authors · 2021-11-23
> **Response to Reviewer PRhm (Part 1 of 3)**
>
> We thank Reviewer PRhm for the suggestive and critical comments, which help us clarify and strengthen our work. Please see our response to each below.
>
> **Q1**: "The use of CLIP which has been trained on human-generated alt-text data goes against the guidelines of the benchmark. This makes the method proposed in the paper not comparable to existing methods in my opinion. The authors should have used the nocaps (XD) for evaluation."
>
> **A1**: We thank the reviewer for raising the issue of NOC (novel object captioning) guidelines. As stated in [1], additional paired image-caption data and ground truth object annotations are prohibited during training. In our work, a pre-trained CLIP is utilized in our framework. It is served as a cross-modal association model, aligning image-caption data and thus rewarding the desirable captions. In recent works of [2,4], pre-trained novel object detectors (using images annotations of Open Images dataset) are also applied for training their NOC models. We would like to point out that, neither [2, 4] or us access additional image-caption pair data or object annotation data during the training stage. Moreover, CLIP only associates image and caption data at the *instance* level during its pretraining, it does not observe any word-level information or object annotation (as [2, 4] did). Thus. we believe we (and [2, 4]) follow the protocols for NOC and thus the comparisons would be fair.
>
> Nevertheless, we are more than happy to provide results on the nocaps (XD) benchmark as suggested. With Conceptual Captions (CC) as an additional dataset (3.1M image-caption pairs) to train our NOC model, and we summarize the results on the nocaps test set below in the table:
>
> |                           | CIDEr  | SPICE |
> | --------------------------|:------:|:-----:|
> | UpDown [1]                | 73.09  | 11.20 |
> | VIVO [2]                  | 100.12 | 14.04 |
> | SimVLM_base [3] | 94.80  | 13.10 |
> | Ours                      | **102.39** | **14.71** |
>
> Note that both VIVO, SimVLM and our method adopt BERT-based backbones for captioning. We note that, VIVO [2] did not specify the details of datasets additionally applied, while SimVLM specifically pointed out that a much larger web-scale dataset (1.8B image-text pairs) was utilized. From the above table, we see that our method still performs favorably against SOTAs on the suggested protocol and benchmark.
>
> We have uploaded our results on the nocaps (XD) benchmark, and here is the link to the leaderboard: https://eval.ai/web/challenges/challenge-page/464/leaderboard/1300. We have also updated the above results in Table 7 with discussions in Appendix B.3 of our revised manuscript.

---

> > ### Author Response · Authors · 2021-11-23
> > **Response to Reviewer PRhm (Part 2 of 3)**
> >
> > **Q2**: "I would have liked to see if you can replace CLIP with other approaches (ViLBERT, VIFIDEL) which also outputs a score indicating how well the caption is aligned with the image. This would form a fair comparison."
> >
> > **A2**: We thank the reviewer for giving us the opportunity to improve our paper. When training our NOC model, we have CLIP associate proper image-caption pair, enforcing caption fidelity and adequacy (i.e., semantic correctness). While it is possible to replace CLIP with ViLBERT, the resulting computational cost would be a major concern, as we discuss below.
> >
> > As noted by the reviewer, one of the objectives for image captioning models is to assess the alignment between an image-caption pair. Considering a training batch with *N* image-caption pairs, the use of CLIP simply calculates the dot products between the features of each possible image-caption pairs to output all *N^2* similarity scores in one forward pass. As for ViLBERT, it deploys a heavy neural network module for performing cross-attention between an image and each possible caption in one forward pass. Thus, for the training batch of the same size, it would require *N* passes to derive all *N^2* similarity scores. While it is possible to utilize ViLBERT for performing such alignments, the computation cost would be significantly higher than the use of CLIP. In our experiment (8 V100 GPUs each with 32 GB memory with *N* = 256), an optimization step for updating our model using CLIP costs 1.8 seconds, and that using ViLBERT would take 256 times longer (i.e., 460 seconds per step). Thus, the use of ViLBERT would make the learning process computationally infeasible.
> >
> > As suggested by the reviewer, another possible replacement of CLIP is VIFIDEL. We also conduct the experiments using VIFIDEL as the association model for comparisons. With the results presented in the following table, we see that the use of VIFIDEL was not able to achieve comparable results as that of CLIP. We believe that this is because VIFIDEL only associates image-caption data using word embeddings of particular object labels, while CLIP assesses such cross-modal data in the instance level, i.e., taking the complete caption of an image into consideration. From the above explanations and experiments, the use of CLIP in our proposed model can be verified.
> >
> > |                |   CIDEr   |   SPICE   |
> > | -------------- |:---------:|:---------:|
> > | Ours (VIFIDEL) |   55.32   |   10.73   |
> > | Ours (CLIP)    | **96.25** | **14.10** |

---

> > > ### Author Response · Authors · 2021-11-23
> > > **Response to Reviewer PRhm (Part 3 of 3)**
> > >
> > > **Q3**: "Since CLIP is trained on alt-text data for images, it would be interesting to see if joint-training on Conceptual Captions (or a similar web-scale data) works better instead."
> > >
> > > **A3**: We thank the reviewer for the suggestion, and we are happy to provide additional results. As discussed in Q1, we follow the suggestion and take the Conceptual Captions (CC) dataset as additional image-caption data (x_l, y_l) to train our NOC model. In the first row of the following table, we list the scores of our proposed NOC model on the nocaps validation set, as shown in Table 1 in the original manuscript. As for the bottom row of the table below, we show our results with additional Conceptual Captions data observed during training. As expected, using additional image-caption training data would improve the NOC performance. It is worth repeating that, as discussed in Q1, SimVLM considered a much larger web-scale dataset (1.8B image-text pairs v.s. 3.1M in CC dataset) but was not able to achieve comparable results as ours did. Thus, the design and learning strategies of our proposed can be successfully verified.
> > >
> > > |           | in domain  |           | near domain |           | out domain |           |  overall   |           |
> > > | --------- |:----------:|:---------:|:-----------:|:---------:|:----------:|:---------:|:----------:|:---------:|
> > > |           | **CIDEr**  | **SPICE** |  **CIDEr**  | **SPICE** | **CIDEr**  | **SPICE** | **CIDEr**  | **SPICE** |
> > > | Ours      |   102.77   |   14.83   |    97.90    |   14.40   |   86.33    |   12.54   |   96.25    |   14.10   |
> > > | Ours(+CC) | **110.56** | **15.23** | **105.16**  | **14.81** | **96.22**  | **13.19** | **104.12** | **14.55** |
> > >
> > > [1] Agrawal et al., "nocaps: novel object captioning at scale", ICCV 2019.
> > >
> > > [2] Hu et al., "VIVO: Visual Vocabulary Pre-Training for Novel Object Captioning", AAAI 2021.
> > >
> > > [3] Wang et al., "SimVLM: Simple Visual Language Model Pretraining with Weak Supervision", Arxiv 2021.
> > >
> > > [4] Zhang et al., "VinVL: Revisiting Visual Representations in Vision-Language Models", CVPR 2021.

---

### Author Response · Authors · 2021-11-23
**General Response**

We thank all the reviewers for the thoughtful feedback and critical comments. We are happy to address the concerns of the reviewers, and revise our manuscript accordingly. We now clarify the issues raised by each reviewer in the following responses, and the revised manuscript has also been submitted.

---

### Decision · Program_Chairs · 2022-01-20

**Decision:**

Reject

**Comment:**

This paper proposes a framework for novel object captioning by combining BERT and CLIP.  The model improves fluency, fidelity, and adequacy of generated captions. However, as reviewers mentioned, the novelty is limited, combining large models and big data to solve a downstream task does not make useful insights at this moment.